# SCaR: Refining Skill Chaining for Long-Horizon Robotic Manipulation via Dual Regularization

**Zixuan Chen**[1]    **Ze Ji**[2]    **Jing Huo**[1]*    **Yang Gao**[1]

[1]State Key Laboratory for Novel Software Technology, Nanjing University, China,
[2]School of Engineering, Cardiff University, UK,
chenzx@nju.edu.cn, jiz1@cardiff.ac.uk, huojing@nju.edu.cn, gaoy@nju.edu.cn

## Abstract

Long-horizon robotic manipulation tasks typically involve a series of interrelated sub-tasks spanning multiple execution stages. Skill chaining offers a feasible solution for these tasks by pre-training the skills for each sub-task and linking them sequentially. However, imperfections in skill learning or disturbances during execution can lead to the accumulation of errors in skill chaining process, resulting in execution failures. In this paper, we investigate how to achieve stable and smooth skill chaining for long-horizon robotic manipulation tasks. Specifically, we propose a novel skill chaining framework called **S**kill **C**haining via D**ua**l **R**egularization (**SCaR**). This framework applies dual regularization to sub-task skill pre-training and fine-tuning, which not only enhances the *intra-skill dependencies* within each sub-task skill but also reinforces the *inter-skill dependencies* between sequential sub-task skills, thus ensuring smooth skill chaining and stable long-horizon execution. We evaluate the SCaR framework on two representative long-horizon robotic manipulation simulation benchmarks: IKEA furniture assembly and kitchen organization. Additionally, we conduct a simple real-world validation in tabletop robot pick-and-place tasks. The experimental results show that, with the support of SCaR, the robot achieves a higher success rate in long-horizon tasks compared to relevant baselines and demonstrates greater robustness to perturbations.

## 1 Introduction

Long-horizon robotic manipulation tasks are characterized by sequences of diverse and interdependent sub-tasks, which makes it crucial to maintain the stability of multi-stage sequential execution. For instance, in the robotic assembly of a stool (Fig. 1) involving two sub-tasks of leg installation, overall success is evaluated based on both the sequential installation success and factors affecting the assembly within environmental constraints. Although recent advances in deep reinforcement learning (RL) and imitation learning (IL) show promise in training robots for such complex tasks [1, 2, 3, 4, 5, 6, 7], managing long-horizon tasks with a scratch RL or IL policy remains challenging due to computational demands, extensive exploration, and intricate step dependencies [8, 9]. Skill chaining, which involves decomposing long-horizon tasks into smaller sub-tasks, pre-training skills for each, and executing them sequentially, offers a practical solution [10, 11]. However, as shown in Fig. 1(a)(b), such methods tend to fail when sub-task skills are insufficiently trained or unexpected states arise due to disturbances, especially when applied to high-degree-of-freedom robots performing contact-rich, long-horizon tasks. [12, 13, 14, 15, 16, 17].

In this paper, we argue that the coordination and enhancing of dependencies within and between sub-task skills is necessary for stable and smooth skill chaining of long-horizon robotic manipulation [10].

---

*Corresponding author.

38th Conference on Neural Information Processing Systems (NeurIPS 2024).

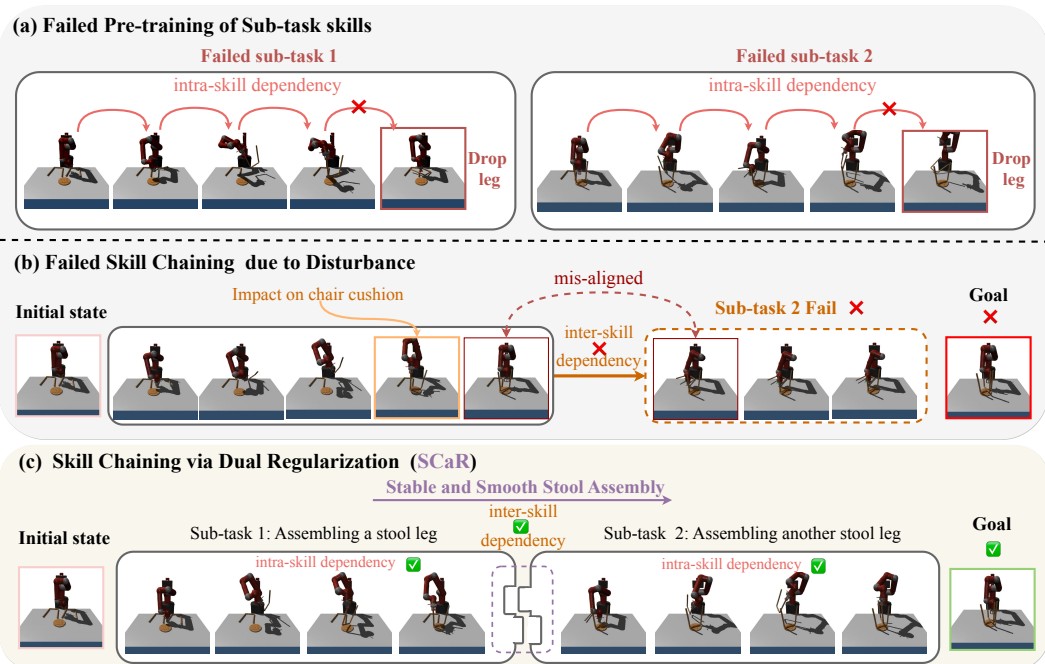

Figure 1: Illustration of the problem setting and the motivation of SCaR, using the example of a stool assembly task with two sub-tasks. Best viewed when zoomed in.

For instance, as depicted in Fig. 1 (a)(b), the robot must consider following two points to ensure the overall task is accomplished: 1) ensuring the gripper consistently grasps and installs the stool leg stably within each sub-task skill range. and 2) ensuring the terminal state of previous skill aligns with the initial state of next skill for smooth skill chaining. We define the above two points as *intra-skill dependencies* between sequential actions within each sub-task skill and *inter-skill dependencies* between sequential sub-task skills, respectively. In this context, we propose a novel robotic skill chaining framework, **S**kill **C**haining via Du**a**l **R**egularization (**SCaR**). This framework enhances the aforementioned dependencies alternately through dual regularization during sub-task skill learning and chaining, aiming to provide stability for the execution of long-horizon robotic manipulation.

Specifically, in the pre-training phase of each sub-task skill, we propose the *adaptive sub-task skill learning* shceme, which employs a two-part policy learning objective that focuses on what sub-tasks the robot should perform (via RL) and how the robot should perform that task (via IL), and utilizes a novel adaptive equilibrium scheduling (AES) regularization to balance these two parts based on the robot's learning progress. This process aims to reinforce the *intra-skill dependencies*, ensuring a coherent sequence of actions in each sub-task skill. Subsequently, *bi-directional adversarial learning* is introduced in the fine-tuning phase of SCaR for better chaining sequential sub-task skills. This mechanism uses bi-directional regularization to bring the terminal state of the current skill close to the initial state of its successor, and also to bring the initial state of the successor close to the terminal state of the current skill. This bi-directional alignment aims to reinforce robust *inter-skill dependencies* between sequential skills. Through the two innovative designs described, SCaR ensures coordination between the *intra-skill* and *inter-skill* dependencies, provides dual constraints for skill learning and skill chaining, as described in Fig. 1 (c), leading to a smooth skill chaining from the inside (**within the sub-task skills**) to the outside (**between sub-task skills**). Experimental results show that compared to scratch-training and skill chaining baselines, SCaR provides better task execution performance and stronger robustness to environmental perturbations in various long-horizon and contact-rich robotic manipulation simulation tasks. In addition, we conduct a simple validation in real-world tabletop robot pick-and-place tasks, and the results show that SCaR achieves a higher task success rate compared to previous skill-chaining methods.

The principal contributions of our work are delineated as follows: **1)** We propose a novel robotic skill chaining framework via dual regularization, SCaR, for smoothly executing long-horizon manipulation

tasks. **2)** We introduce an adaptive sub-task skill learning scheme that acts as a regularization to enhance *intra-skill dependencies* between sequential actions within each sub-task skill. **3)** We develop a bi-directional adversarial learning mechanism that serves as a regularization for reinforcing *inter-skill dependencies* between sequential sub-task skills. **4)** In all eight simulated long-horizon robotic manipulation tasks and simple real-world pick-and-place tasks, SCaR demonstrates significantly better performance than scratch-training and skill-chaining baselines. Video demonstrations are available at: `https://sites.google.com/view/scar8297`.

## 2 Related Work

### 2.1 Long-horizon Robotic Manipulation

Training robots from scratch for complex, long-horizon tasks using reinforcement learning (RL) and imitation learning (IL) is challenging due to computational demands and distributional errors. Solutions involve decomposing tasks into reusable sub-tasks [18]. Typically, such algorithms consist of a set of sub-policies that can be obtained through various methods, such as unsupervised exploration [19, 20, 21, 22, 23], learning from demonstrations [5, 6, 24, 25], and predefined measures [26, 27, 28, 29, 14]. Despite the merits of each of these approaches, they do not address well the challenges of long-horizon robot manipulation in environments that are object-rich, contact-rich, and characterized by multi-stage tasks [28, 29, 14]. Thus, even when pre-trained skills are provided, ensuring a smooth connection between manipulation policies remains a formidable challenge.

### 2.2 Skill Chaining for Long-horizon Tasks

Previous skill chaining methods for long-horizon tasks mainly focus on updating each sub-task policy to encompass the terminal state of the previous policy [11, 14, 30], implementing option chains [11, 31, 32] to forge logical skill sequences, or utilizing modulated skills to facilitate smoother transitions [33, 34, 35, 36, 14, 16]. However, these methods, while effective, often lead to a broad range of skill start and end states, a challenge in complex robotic manipulation tasks. T-STAR [15] is closely related to our work, addressing this by regularizing the learning process with a discriminator to control the expansion of the terminal state space. However, it focuses only on uni-directional dependencies between skills and ignores intra-skill dependencies within sub-task skills under long-horizon goals. Sequential Dexterity [17] centers on dexterous hand manipulation, introducing an optimization process to backpropagate long-term rewards across a policy chain. However, its scope still primarily emphasizes strengthening the dependencies between sub-task skills. GSC [37] attempts to solve skill chaining by employing diffusion models. It trains and chains primitive skills (pick, place, push, pull) through a Transformer-based skill diffusion model. However, due to the use of Transformer-based techniques, GSC requires high computational resources and cannot scale well to task environments with object-rich and contact-rich conditions. Our method instead employs simple and intuitive dual regularization constraints based on the lightweight policy network. By coordinating the dependencies within and between skills, we achieve refinement within sub-task policies and bi-directional alignment between them. This allows for stable skill chaining while also being scalable to various long-horizon manipulation tasks.

## 3 Preliminaries

Among several related works on skill chaining, we consider a challenging yet practical problem setting that *deals with long-horizon manipulation tasks through a combination of reinforcement learning (RL) and imitation learning (IL)*. In each sub-task in the long-horizon task, we consider robotic agents acting within a finite-horizon Markov Decision Process [38] $(\mathcal{S}, \mathcal{A}, \mathcal{P}, r, \gamma, d_\mathcal{I}, T)$, where $\mathcal{S}$ is the state space, $\mathcal{A}$ is the action space, $\mathcal{P}(s'|s, a)$ is the transition function, $r(s, a, s')$ is the reward function, $\gamma$ is the discount factor, $d_\mathcal{I}$ is the initial state distribution, and $T$ is the episode horizon of sub-task. We define a policy $\pi : \mathcal{S} \to \mathcal{A}$ that maps states to actions and correspondingly moves the robotic agent to a new state according to the transition probabilities. This sub-task policy is trained to maximize the expected sum of discounted rewards $\mathbb{E}_{(s,a)\sim\pi}[\sum_{t=1}^{T} \gamma^t r(s_t, a_t, s_{t+1})]$. We assume that each sub-task policy has an initial state set $\mathcal{I} \in \mathcal{S}$ and a terminal state set $\beta \in \mathcal{S}$, where the initial set $\mathcal{I}$ contains all the initial states that lead to the successful execution of the policy and the terminal state set $\beta$ contains all the final states of the successful execution. The environment

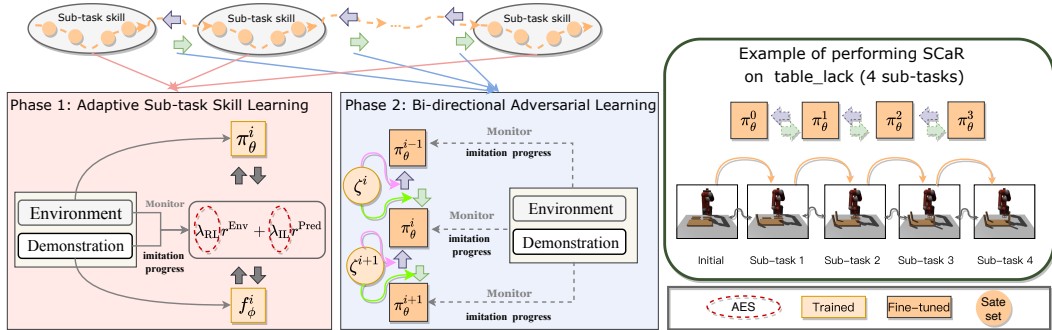

Figure 2: The Pipeline of **S**kill **C**haining via D**u**al **R**egularization (SCaR). **(Left) Phase 1**: Sub-task skill pre-training ( ) merges environmental feedback and expert guidance, using adaptive equilibrium scheduling (AES) regularization to balance learning, which enhances intra-skill dependencies within skills. **(Middle) Phase 2**: Bi-directional discriminators ( ) coupled with AES to fine-tune pre-trained sub-task skills, as regularization for reinforcing inter-skill dependencies. **(Right) Evaluation**: Evaluation of SCaR on long-horizon manipulation.

provides the environmental feedback for each step taken by the agent and success metrics for each sub-task, derived from the terminal states of sub-task policy. For instance, as shown in Fig. 1(c), the alignment of the back and legs of the stool triggers the connect action and the realization of the sub-task goal, which indicates the successful completion of the sub-task. Additionally, we posit that during each sub-task policy learning, the agent receives a set of pre-defined expert demonstrations, $\mathbb{D}^E = \{\tau_1^E, \ldots, \tau_N^E\}$, to facilitate the IL process. Here, $N$ represents the number of episodes, and each demonstration comprises a sequence of state-action pairs, $\tau^E = (s_1, a_1, \ldots, s_{T-1}, a_{T-1}, s_T)$.

# 4 Method

In Section 4.1, we present the pipeline of the **SCaR** framework. Sections 4.2 and 4.3 provide further elaboration on the key design elements.

## 4.1 Overall Pipeline

As illustrated in Fig. 2, the **SCaR** framework has two phases: **(a) pre-training (adaptive sub-task skill learning) and (b) fine-tuning (bi-directional adversarial learning)**. In the pre-training phase, the agent co-learns sub-task skills by integrating environmental feedback and expert demonstrations. In the fine-tuning phase, it refines these skills through bi-directional adversarial learning, enabling sequential integration of sub-task skills. After fine-tuning, SCaR can smoothly chain sub-task skills to complete long-horizon manipulation tasks. Specific modules and mechanisms for these phases are detailed in Sections 4.2 and 4.3.

## 4.2 Adaptive Sub-task Skill Learning

**Weighted Reward Function** To learn sub-task skills better, we combine goal-conditional RL and generative adversarial imitation learning (GAIL) [39], to pre-train skills that enable the agent to perform challenging sub-tasks in a desired expert behavioral style [40, 15]. More specifically, we consider the weighted reward function that is used to train each sub-task policy $\pi_i^\theta$ consists of two components specifying: *what sub-task the agent should perform* - learning from environmental feedback, and 2) *how the agent should perform that task* - learning from expert demonstrations:

$$r(s_t, a_t, s_{t+1}; \phi) = \lambda_{\mathrm{RL}} r_i^{\mathrm{Env}}(s_t, a_t, s_{t+1}, g) + \lambda_{\mathrm{IL}} r_i^{\mathrm{Pred}}(s_t, a_t; \phi). \tag{1}$$

As shown in Eq. 1, the first component is represented by a task-specific reward $r_i^{\mathrm{Env}}(s_t, a_t, s_{t+1}, g)$, which defines general objectives that the agent should satisfy to fulfill a given sub-task goal $g$ for current MDP $\mathcal{M}$ (e.g. assembling a stool leg). The second component is represented through a learned task-agnostic predict-reward $r_i^{\mathrm{Pred}}(s_t, a_t; \phi)$, which specifies manipulation details of the behaviors that the agent should adopt when performing the sub-task (e.g., the expert way to grab a stool leg

and attach it), and $r_i^{\mathrm{Pred}}(s_t, a_t; \phi)$ is the predicted reward by a least-square GAIL discriminator $f_\phi^i$ [41, 40, 15], which is more stable than the standard GAIL objective using the sigmoid cross-entropy loss function. Therefore, the predicted reward is:

$$r_i^{\mathrm{Pred}}(s_t, a_t; \phi) = \max[0, 1 - 0.25 \cdot [f_\phi^i(s_t, a_t) - 1]^2]. \tag{2}$$

We adopt the training objective of the least-squares GAIL discriminator [41] with a gradient penalty term [42, 43], This penalty term mitigates the instability of the training dynamics due to the interplay between the discriminator and the policy [40], as follows:

$$\mathrm{argmin}_{f_\phi^i} \mathbb{E}_{(s)\sim\mathbb{D}^E}[(f_\phi^i(s) - 1)^2] \quad + \mathbb{E}_{(s)\sim\pi_\theta^i}[(f_\phi^i(s) + 1)^2] + \frac{\eta^{\mathrm{gp}}}{2}\mathbb{E}_{(s)\sim\mathbb{D}^E}[\|\nabla_s f_\phi^i(s)\|^2], \tag{3}$$

where $\eta^{\mathrm{gp}}$ is a manually-specified coefficient. The scales of $r^{\mathrm{Env}}$ and $r^{\mathrm{Pred}}$ in previous related works are set by fixed weights and linearly combined into the final reward function [40, 15]. This could lead to the agent rigidly imitating experts and curbing self-exploration, finding it difficult to adjust intra-skill dependencies and adapt to dynamic task perturbations. We propose a principle to counter this: ***If the agent fails to imitate the expert's demonstration well, it should shift focus to self-learning from the environment. Conversely, effective imitation should continue, focusing on the expert to mitigate low sample efficiency in reinforcement learning.*** Accordingly, we extend the automatic discount scheduling (ADS) solution [9] to our problem setting, and propose adaptive equilibrium scheduling (AES) to regularize the scales of $r^{\mathrm{Env}}$ and $r^{\mathrm{Pred}}$ in sub-task skill learning for adaptive scheduling the focus of reinforcement and imitation learning, as shown in Fig. 3.

**Adaptive Equilibrium Scheduling (AES) Regularization** Specifically, AES balances the scales of $r^{\mathrm{Env}}$ and $r^{\mathrm{Pred}}$ during the learning process of each skill through adaptive scheduling of $\lambda_{\mathrm{RL}}$ and $\lambda_{\mathrm{IL}}$, according to how well the agent imitates the expert's demonstration. To capture the agent's imitation progress, AES refers to the solution in ADS [9] and uses the imitation identifier $\Phi$ to continuously monitor whether the agent is imitating the expert demonstration well enough.

At the beginning of training, the agent is assigned two initial balance factors $\lambda_{\mathrm{RL}} = \alpha, \lambda_{\mathrm{IL}} = 1 - \alpha$, where base exponent $\alpha \in [0, 1]$. We set $\alpha = 0.5$ in the experiments and the agent is assigned two identical balance factors $\lambda_{\mathrm{RL}} = \lambda_{\mathrm{IL}} = 0.5^2$, indicating that at the beginning of learning, the agent imitates the expert's behavior with the same weight as the behavior of environment exploration according to the task goal. As training progresses, the imitation progress recognizer $\Phi$ is queried periodically to monitor the progress of the agent's imitation of the expert's behavior. $\Phi$ receives the agent's collected trajectories and infers the agent's current imitation progress $p \in [0, T)$, where $p$ in an integer and $T$ is the step of the entire episode.

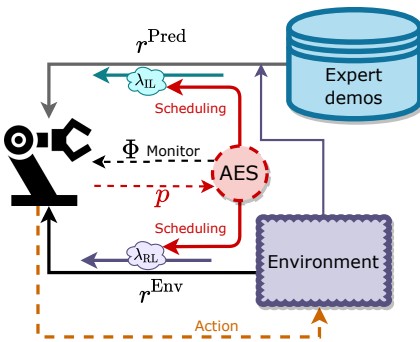

Figure 3: AES regularization for sub-task skill learning.

The construction of $\Phi$, with reference to ADS, first requires the construction of a sequence $\mathbf{Q}(q_1, \ldots, q_T)$, where $q_i = \mathrm{argmin}_j c(s_i, s_j^E)$ is the index of the nearest neighbor of $s_i$ in $\tau^E$, $c$ is the cosine similarity. The progress alignment between $\tau$ and $\tau_j^E$ is measured as the length of the longest increasing subsequence (LIS) in $\mathbf{Q}$, denoted as $LIS(\tau, \tau^E)$. Specifically, the agent's imitation progress $p$ is increased by 1 if the following inequality holds:

$$\max_{\acute{\tau}^E \in \mathbb{D}^E} LIS(\tau_{1:p+1}, \acute{\tau}_{1:p+1}^E) \geq \rho \times \min_{\acute{\tau}^E, \grave{\tau}^E \in \mathbb{D}^E} LIS(\acute{\tau}_{1:p+1}^E, \grave{\tau}_{1:p+1}^E), \tag{4}$$

where $\acute{\tau}^E \neq \grave{\tau}^E$, the subscript $1:p+1$ denotes the first $p+1$ steps of the trajectory, and $\rho \in [0, 1]$ controls the strictness of the imitation progress monitoring. This suggests that the similarity of the agent trajectory to its best matching expert trajectory at time step $p+1$ exceeds the minimal similarity criterion within the expert demonstration. See Appendix B for detailed explanation of AES.

After obtaining the current imitation progress $p$ of the agent, AES then adopts a mapping function $\varphi_\lambda(p)$ to schedule the two new balance discount factors $\lambda_{\mathrm{RL}}$ and $\lambda_{\mathrm{IL}}$. Straightforward idea of setting

---

[2]We further explore what effect different $\alpha$ would have in the Ablation Experiments.

$\varphi_\lambda(p)$ is that **If $p$ is larger and reaches a certain threshold, i.e., the agent is able to imitate the expert behavior well, then the more the agent tends to imitate the expert's behavior in subsequent training, and vice versa.** Therefore, we set the threshold as $\frac{T}{2}$. If $p \in [0, \frac{T}{2})$, we propose $\varphi_\lambda(p) = 1 - e^{\left(-\frac{p}{k}\right)}$; if $p \in [\frac{T}{2}, T)$, we propose $\varphi_\lambda(p) = e^{\left(-\frac{p - \frac{T}{2}}{k}\right)}$, where $k$ is used to flatten the curve of the mapping function. Then $\lambda_{\text{RL}}$ and $\lambda_{\text{IL}}$ are scheduled to be :

$$\begin{cases} \lambda_{\text{RL}} = \alpha^{\varphi_\lambda(p)}, \lambda_{\text{IL}} = 1 - \alpha^{\varphi_\lambda(p)}. & \text{if } p \in [0, \frac{T}{2}) \\ \lambda_{\text{IL}} = \alpha^{\varphi_\lambda(p)}, \lambda_{\text{RL}} = 1 - \alpha^{\varphi_\lambda(p)}. & \text{if } p \in [\frac{T}{2}, T) \end{cases} \tag{5}$$

Consequently, the RL and IL components of sub-task skill learning can be adaptively scheduled and regularized through AES, effectively enhancing *intra-skill dependencies* between sequential actions. The pseudo-code of adaptive sub-task skill learning is outlined in Algorithm 1 in Appendix A.1.

### 4.3 Bi-directional Adversarial Learning for Skill Chaining

Executing pre-trained sub-task skills sequentially without considering inter-skill dependencies may lead to failure. To address this, we propose bi-directional adversarial learning to further refine and better integrate sequential sub-task skills. The pseudo-code of bi-directional adversarial learning is outlined in Algorithm 2 in Appendix A.2.

**Bi-directional Regularization** In contrast to previous uni-directional regularization schemes that only augment the initial state set $\mathcal{I}_i$ or regularize the terminal state set $\beta_i$ [12, 15], we impose the *bi-directional constraints* ($\mathcal{C}_1, \mathcal{C}_2$) on inter-skill dependencies, facilitating smooth skill chaining, as shown in Fig 4. With the bi-directional constraint, we implement the bi-directional adversarial learning, centered on the joint training of a *bi-directional discriminator*, denoted by $\zeta_\omega^i$, which is adept at distinguishing between the terminal state set of the preceding policy and the initial state set of the subsequent policy. The bi-directional constraints $\mathcal{C}_1, \mathcal{C}_2$ are defined as Eq. 10:

$$\begin{aligned} \textbf{next initial} \rightarrow \textbf{current terminal:} \quad & \mathcal{C}_1 = \mathbb{E}_{s_\mathcal{I} \sim \mathcal{I}_{i+1}}[\zeta_{\omega_1}^i(s_\mathcal{I}) - 1]^2 + \mathbb{E}_{s_T \sim \beta_i}[\zeta_{\omega_1}^i(s_T)]^2 \\ \textbf{previous terminal} \rightarrow \textbf{current initial:} \quad & \mathcal{C}_2 = \mathbb{E}_{s_T \sim \beta_{i-1}}[\zeta_{\omega_2}^i(s_T) - 1]^2 + \mathbb{E}_{s_\mathcal{I} \sim \mathcal{I}_i}[\zeta_{\omega_2}^i(s_\mathcal{I})]^2 \end{aligned} \tag{6}$$

$\zeta_{\omega_1}^i$ and $\zeta_{\omega_2}^i$ are two separate networks, each used to minimize the adversarial learning process in two different directions, and the parameters of the two networks are averaged and combined into $\zeta_\omega^i$. In summary, $\zeta_\omega^i$ is trained for each policy to minimize the objective function[3]: $\mathcal{L}_i(\omega) = \frac{1}{2}\mathcal{C}_1 + \frac{1}{2}\mathcal{C}_2$. Guided by $\zeta_\omega^i$, the bi-directional adversarial learning not only steers the terminal state set of the current policy towards the initial state set of the subsequent policy, but also ensures alignment of the initial state set of the subsequent policy with the terminal state set of current policy. This dual alignment establishes a balanced mapping between the initial and terminal states of sequential skills to reinforce inter-skill dependencies, ensure consistency and stability in multistage tasks, and guarantee smooth transitions between sequential skills. Accordingly, the *bi-directional regularization* can be added to the overall objective function of policy learning in the form of the following reward term: $r_i^{\text{Bi}}(s; \omega) = \mathbb{1}_{s \in \beta_i} \zeta^{i+1}(s) + \mathbb{1}_{s \in \mathcal{I}_i} \zeta^{i-1}(s)$.

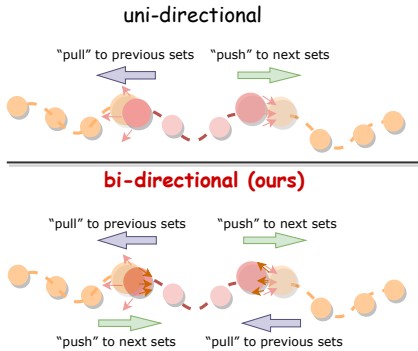

Figure 4: Bi-directional regularization for sub-task skill chaining.

**Overall Objective Function** So far, the objective function via dual regularization, i.e., AES regularization and bi-directional regularization, to pre-train, fine-tune and chain sub-task skills can be rewritten as a weighted sum of the individual reward terms:

$$r_i(s_t, a_t, s_{t+1}; \phi) = \underbrace{\lambda_{\text{RL}} r_i^{\text{Env}}(s_t, a_t, s_{t+1}, g) + \lambda_{\text{IL}} r_i^{\text{Pred}}(s_t, a_t; \phi)}_{\textbf{AES regularization}} + \underbrace{\lambda_{\text{Bi}} r_i^{\text{Bi}}(s_{t+1}; \omega)}_{\textbf{bi-directional regularization}}, \tag{7}$$

---

[3]We explore the impact of different scales of $\mathcal{C}_1$ and $\mathcal{C}_2$ in Appendix D.3

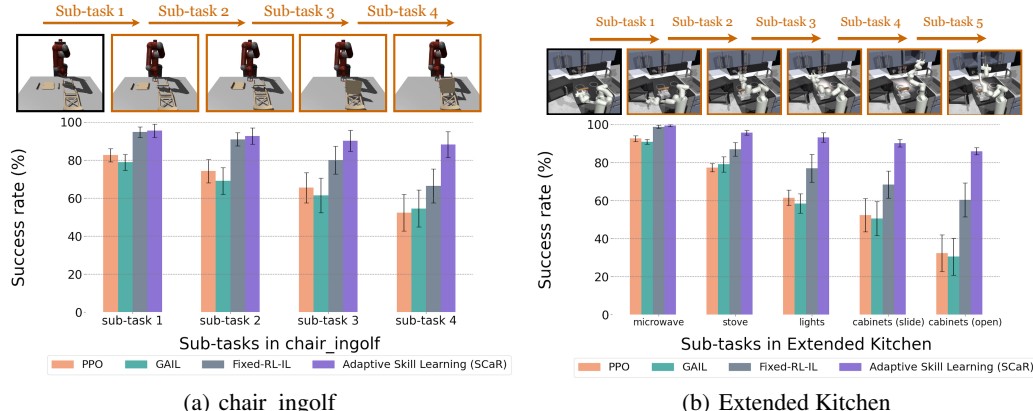

Figure 5: Evaluation Performance of Sub-task Skill Learning. Best viewed zoomed.

where $\lambda_{\mathrm{Re}}$ is the weighting factor of the bi-directional regularization. The objective function features AES regularization and bi-directional regularization to enhance intra- and inter-skill dependencies. It enables the agent to adaptively pre-train skills that can solve different sub-tasks well through environmental feedback and expert guidance, and further fine-tune them through the bi-directional discriminator to achieve dual alignment between sequential skills. At the same time, the fine-tuned sub-task skills help to collect terminal and initial states to refine the bi-directional discriminator. This iterative process ensures smooth long-horizon task skill chaining.

# 5 Experiments

## 5.1 Experiment Setup

We conduct simulation experiments on six IKEA furniture assembly tasks and two kitchen organization tasks, and also perform long-horizon pick-and-place experiments on the real Sagittarius K1 robot. Please refer to the Appendix for more detailed simulation experiment setup (Appendix G), network architecture (Appendix H), training details (Appendix I), more quantitative (Appendix D) and qualitative results (Appendix E) of the simulation tasks, and the real-robot experiments (Appendix F).

**Furniture Assembly**   We conduct experiments in six IKEA furniture assembly tasks in [44]: *chair_agne*, *chair_bernhard*, *chair_ingolf*, *table_lack*, *toy_table*, and *table_dockstra*.

1) *chair_agne*: Two stool legs need to be picked up and aligned with the cross notches on the stool back. 2) *chair_bernhard*: The two chair supports need to be taken and aligned with the slots at the bottom of the chair surface. 3) *chair_ingolf*: Two chair supports and front legs need to be attached to the chair seat, which  must then be secured to the chair back while avoiding collision with each other. 4) *table_lack*: The four table legs need to be picked up and aligned with the corners of the tabletop. 5) *toy_table*: The four table legs need to be picked up and aimed and inserted with the four notches on the table back. 6) *table_dockstra*: After supporting the two bases with table leg, the table top needs to be mounted while preventing collision. For each assembly task, we define the assembly of individual parts as sub-tasks. We collect 200 demonstrations per sub-task using a procedural assembly policy for imitation learning. Each demonstration consists of 150 steps.

**Kitchen Organization**   We use the Franka Kitchen tasks in D4RL [45] and collect 200 demonstrations per sub-task for imitation learning. Specifically, we refer to the kitchen task in  [46] and further extend the task sequence: in the **Kitchen task**, the 7-DoF Franka Emika Panda arm needs to perform 4 sequential sub-tasks, namely *Turn on the microwave - Move the kettle - Turn on the stove - Turn on the light*. In the **Extended Kitchen task**, the robot needs to perform 5 sequential sub-tasks: *Turn on the microwave - Turn on the stove - Turn on the light - Slide the cabinet to the right - Open the cabinet*, in which the sub-tasks have a lower probability of switching and is more challenging.

**Baselines**   We compare SCaR with the following two types of baselines:

Table 1: Long-horizon tasks execution performance (varies by sub-task completion progress): *tasks with 2 sub-tasks progress by 0.5 per sub-task, *tasks with 4 sub-tasks by 0.25, *tasks with 5 sub-tasks by 0.2, and table_dockstra with 3 sub-tasks by 0.3, where 0.9 indicates completion of all tasks. Best viewed zoomed.

| Method | Furniture Assembly | | | | | | | Kitchen Organization | | |
| --- | --- | --- | --- | --- | --- | --- | --- | --- | --- | --- |
| | chair_agne | chair_bernhard | chair_ingolf | table_lack | toy_table | table_dockstra | All | Kitchen | E-Kitchen | All |
| PPO (Scratch RL) | 0.54±0.18 | 0.42±0.12 | 0.14±0.03 | 0.09±0.01 | 0.00±0.00 | 0.31±0.12 | 0.25±0.15 | 0.13±0.05 | 0.03±0.00 | 0.08±0.04 |
| GAIL (Scratch IL) | 0.31±0.05 | 0.23±0.02 | 0.00±0.00 | 0.00±0.00 | 0.00±0.00 | 0.21±0.04 | 0.12±0.09 | 0.00±0.00 | 0.00±0.00 | 0.00±0.00 |
| Fixed-RL-IL | 0.68±0.12 | 0.53±0.07 | 0.22±0.08 | 0.21±0.11 | 0.13±0.02 | 0.43±0.07 | 0.37±0.15 | 0.33±0.06 | 0.18±0.02 | 0.26±0.06 |
| SkiMo | 0.75±0.09 | 0.62±0.05 | 0.47±0.03 | 0.58±0.14 | 0.34±0.06 | 0.62±0.11 | 0.56±0.11 | 0.57±0.08 | 0.21±0.04 | 0.39±0.13 |
| Policy Sequencing | 0.89±0.08 | 0.82±0.09 | 0.77±0.12 | 0.63±0.28 | 0.45±0.18 | 0.61±0.14 | 0.70±0.16 | 0.53±0.11 | 0.36±0.09 | 0.44±0.09 |
| T-STAR | 0.92±0.02 | 0.90±0.04 | 0.89±0.04 | 0.90±0.07 | 0.71±0.21 | 0.77±0.09 | 0.85±0.09 | 0.68±0.13 | 0.48±0.08 | 0.58±0.10 |
| SCaR w/o Bi | 0.93±0.04 | 0.92±0.02 | 0.91±0.01 | 0.93±0.02 | 0.80±0.10 | 0.79±0.02 | 0.88±0.05 | 0.75±0.08 | 0.57±0.14 | 0.66±0.09 |
| SCaR w/o AES | 0.95±0.03 | **0.94**±0.03 | 0.93±0.02 | 0.95±0.04 | 0.85±0.06 | 0.80±0.03 | 0.91±0.05 | 0.77±0.07 | 0.61±0.13 | 0.74±0.05 |
| **SCaR (Ours)** | **0.98**±0.02 | **0.96**±0.04 | **0.95**±0.03 | **0.97**±0.03 | **0.92**±0.05 | **0.88**±0.02 | **0.94**±0.03 (12% ↑) | **0.84**±0.16 | **0.73**±0.17 | **0.78**±0.12 (18% ↑) |

**Scratch Training:** 1) **PPO** is a model-free RL algorithm [47] that utilizes environmental rewards to learn tasks from scratch. 2) **GAIL** [39] is an adversarial imitation learning method to learn tasks from scratch, with a trained discriminator for distinguishing state-action distributions of experts and agents. 3) **Fixed-RL-IL** [40] uses fixed-weight environmental rewards and GAIL rewards to train policies from scratch. 4) **SkiMo** [46] is a model-based hierarchical RL approach that learns dynamic skill models for predicting outcomes in downstream tasks, which is used to test if modularly skill chaining method can surpass model-based scratch-training method on long-horizon tasks.

**Skill Chaining:** 1) **Policy Sequencing** [12] focuses on sequentially expanding the initial sets in skill chaining. 2) **T-STAR** [15] incorporates a discriminator to uni-directionally regularize the terminal states of sub-skills in a skill chaining. 3) **SCaR w/o Bi** reference to T-STAR during the fine-tuning phase, only uni-directional regularization of the terminal state set is performed to verify the validity of the proposed bi-directional regularization. 4) **SCaR w/o AES** fixes the scales of the two reward terms at 0.5 at all times to verify the effectiveness of the proposed AES regularization.

## 5.2 Quantitative Results

**Sub-task Skill Learning Performance** First, we evaluate the proposed adaptive sub-task skill learning scheme in the sub-tasks of furniture assembly and kitchen organization. Specifically, we treat each sub-task as a separate task for policy learning and take the success rate of the trained policy tested in the reset sub-task as the criterion. All methods are trained in each sub-task with 5 random seeds, 150 million environment steps, and evaluated with the average success rate over 100 testing episodes. As shown in the Fig. 5, in *chair_ingolf* and Extended Kitchen tasks, even with the increase of objects in the environment and the increase of unpredictable perturbations, our proposed adaptive skill learning learns good sub-task skills and consistently maintains a task success rate of more than 85% in all stages of the sub-task. In contrast, the PPO (only RL rewards), GAIL (only IL rewards), and Fixed-RL-IL (fixed RL and IL reward weights) baselines fail to maintain good sub-task success rates as the number of sub-task stages increases. This result well validates that our proposed adaptive weighted reward function based on AES regularization enhances *intra-skill dependencies* for multi-stage sub-task learning and brings effectiveness and stability.

**Long-horizon Execution Performance** We then demonstrate the performance of SCaR in performing 8 long-horizon tasks in IKEA furniture assembly and kitchen organization. Table 1 shows the mean and standard deviation for these 8 tasks across 200 testing episodes with 5 different seeds. The PPO and GAIL baselines show minimal success on tasks with 4 and 5 sub-tasks, indicating the difficulty of learning complex multi-stage tasks solely from reward signals or expert demonstrations. The fixed RL-IL baseline, although improved compared to PPO and GAIL, mostly completed only one sub-task, which highlights the limitations of using fixed RL and IL reward weights in long-horizon tasks. While SkiMo achieves better success rates than model-free methods by building dynamic skill models, its performance remains inconsistent on long-horizon tasks due to its scratch learning nature. The performance of these scratch baselines demonstrates the importance of effective staged sub-task learning for long-horizon tasks. The results in Table 1 further highlight the superiority of the SCaR framework. By reinforcing *intra- and inter-skill dependencies*, task success rates are considerably higher than previous skill chaining approaches such as Policy Sequencing and T-STAR, which primar-

ily address uni-directional inter-skill dependencies. Compared to T-STAR, SCaR increases average success rates by more than 12% on six furniture assembly tasks and 18% on two kitchen tasks.[4].

## 5.3 Robustness to Perturbations

Perturbation tests are conducted to evaluate the robustness of skill chaining for two furniture assembly tasks. As shown in the top figure of Table 2, for the *chair_bernhard* task, the perturbation involves applying external joint torque to the robotic arm, moving the chair back before assembling the second support. For the *chair_ingolf* task, the perturbation is applied by exerting external torque on the robotic arms, causing them to move slightly before mounting the assembled chair seat to the chair back. The results in Table 2 highlight the detrimental impact of environmental perturbations on the success rates of baseline methods during the execution of multiple sub-task

Table 2: Comparison of the robustness of skill chaining in perturbed environments.

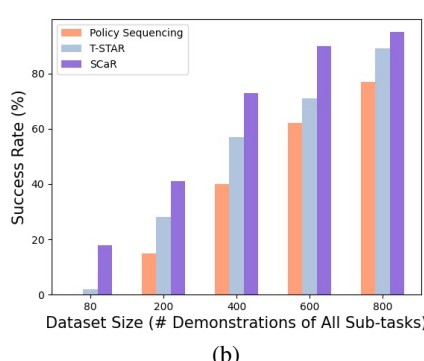

| Method | chair_bernhard | | chair_ingolf | |
|---|---|---|---|---|
| | **No Perturb** | **Perturb** | **No Perturb** | **Perturb** |
| **Policy Sequencing** | $0.82 \pm 0.09$ | $0.51 \pm 0.04$ | $0.77 \pm 0.12$ | $0.50 \pm 0.10$ |
| **T-STAR** | $0.90 \pm 0.04$ | $0.60 \pm 0.08$ | $0.89 \pm 0.04$ | $0.59 \pm 0.04$ |
| **SCaR w/o Bi** | $0.92 \pm 0.02$ | $0.65 \pm 0.11$ | $0.91 \pm 0.01$ | $0.63 \pm 0.05$ |
| **SCaR w/o AES** | $0.94 \pm 0.03$ | $0.74 \pm 0.09$ | $0.93 \pm 0.02$ | $0.71 \pm 0.07$ |
| **SCaR (Ours)** | $\mathbf{0.96} \pm 0.04$ | $\mathbf{0.85} \pm 0.11$ | $\mathbf{0.95} \pm 0.03$ | $\mathbf{0.80} \pm 0.13$ |

skills. Methods like Policy Sequencing and T-STAR, which focus solely on inter-skill dependencies through uni-directional regularization, struggle to complete tasks after perturbations. In contrast, SCaR, demonstrates more robust performance even under unseen perturbations. These results further support the advantages of our proposed *dual regularization* for stable skill chaining on long-horizon manipulation tasks.

## 5.4 Ablations and Analysis

We perform ablation studies to explore the important factors that affect the performance of SCaR.

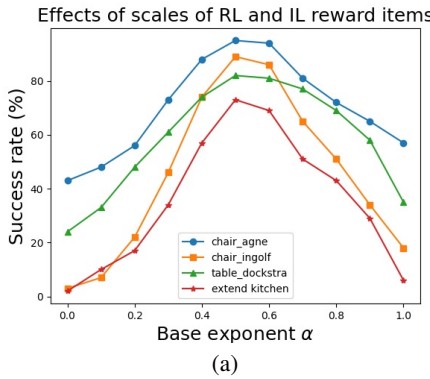

Figure 6: Ablation experiments.

**Modular Ablation** We investigate how the adaptive sub-task skill learning and bi-directional adversarial learning impact skill chaining through SCaR w/o Bi and SCaR w/o AES. As shown in Table 1, without bi-directional regularization, SCaR w/o Bi experiences significant performance drops in tasks with more than two sub-tasks but still outperforms T-STAR. This is because SCaR w/o Bi maintains the adaptive scheduling of AES during sub-task skill learning, underscoring the importance of focusing on the *intra-skill dependencies* between successive actions. Similarly, the absence of AES regularization reduces SCaR w/o AES's performance, though it still maintains stable outcomes. This underscores the importance of reinforcing *inter-skill dependencies* on long-horizon tasks and reaffirms the contribution of bi-directional regularization. As shown in Table 2, SCaR w/o

---

[4]The overall increase is somewhat modest due to averaging the success rates of the 2, 3, and 4 sub-tasks and the 4 and 5 sub-tasks, respectively.

Bi, though slightly more robust than T-STAR due to the presence of AES, still faces challenges in adapting to perturbations and maintaining stable skill chaining because of its uni-directional fine-tuning limitations. SCaR w/o AES manages to maintain a certain level of performance stability under perturbations, thanks to bi-directional regularization, which ensures the bi-directional alignment of initial and terminal states between skills. The results show that the pre-trained skills via AES exhibit enhanced *intra-skill dependencies* within sub-tasks, and bi-directional regularization ensures stable long-horizon execution, even in the presence of perturbations, by reinforcing *inter-skill dependencies*.

**Parametric Ablation** We further investigate the impact of different scales of RL and IL reward terms, as well as the size of expert demonstration datasets. The effect of varying the base exponent $\alpha$ on task success rates is tested across four tasks: *chair_agne*, *chair_ingolf*, *table_dockstra*, and *extend kitchen*. As depicted in Fig. 6(a), SCaR achieves the highest success rates in all four tasks when $\alpha = 0.5$, indicating a balance between RL and IL at the beginning of learning. When $\alpha$ becomes smaller, emphasizing IL at the start, performance decreases more steeply. Conversely, as $\alpha$ becomes larger, giving more weight to RL, performance also declines but at a slower rate. We also evaluate the impact of different sizes of expert datasets on three skill chaining methods: Policy Sequencing, T-STAR, and SCaR, specifically in the *chair_ingolf* task. We vary the overall task expert data size from 80, 120, 200, 400, 600, to 800 demos. As shown in Fig. 6(b), the results indicate significant performance improvement when increasing the dataset size from 400 to 800 demos, while the improvement is less pronounced when going from 80 to 120 demos. This demonstrates the importance of the demo dataset size in the effectiveness of data-driven approaches like skill chaining.

## 6 Discussion

**Limitation and future directions** The primary limitation of our work is that the sub-task division for long-horizon tasks is predefined and does not incorporate visual or semantic processing of objects. Expanding our framework to handle longer-horizon visual manipulation tasks is a direction we aim to explore in future research. For example, integrating a more scalable architecture [48] and performing large-scale pre-training on extensive datasets [49, 50] are promising directions. Another avenue worth exploring is applying our framework to real-world robotic furniture assembly tasks, rather than only staged pick-and-place tasks. Constructing a deployment environment for real-world furniture assembly and ensuring the complete insertion of each furniture module presents significant challenges. We discuss additional limitations and potential solutions in further detail in Appendix J.

**Conclusion** In this paper, we introduce SCaR, a novel skill chaining framework that ensures smooth and stable execution of long-horizon robotic manipulation tasks via dual regularization within and between sub-task skills. Extensive experiments demonstrate that the SCaR framework achieves better task success rates than the baseline methods in both simulated and real-robot manipulation tasks, while being robust against perturbations. We hope this work will inspire future research to further explore the potential of skill chaining for long-horizon robotic manipulation.

## Acknowledgments and Disclosure of Funding

This work was supported in part by the Science and Technology Innovation 2030 New Generation Artificial Intelligence Major Project under Grant 2021ZD0113303; in part by the National Natural Science Foundation of China under Grant 62192783, Grant 62276128; in part by the Collaborative Innovation Center of Novel Software Technology and Industrialization; in part by the Fundamental Research Funds for the Central Universities under Grant 14380128; in part by the Postgraduate Research & Practice Innovation Program of Jiangsu Province under Grant KYCX24_0263.

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

# Technical Appendix

## A  Pseudo-code

Pseudo-code for adaptive sub-task skill learning and bi-directional adversarial learning are shown in Algorithm 1 and Algorithm 2 respectively. We highlight the key differences between our method and the most relevant T-STAR with a gray background.

### A.1  Adaptive Sub-task Skill Learning

As shown in Algorithm 1, the innovation of the sub-task skill learning scheme we propose, compared to previous methods, consists of two parts: 1) We use a more stable weighted reward function for policy learning of sub-task skills, as shown in Eq. 1 and Eq. 3 in the main paper. 2) We introduce AES regularization constraints into this weighted reward function to periodically adaptively schedule the scale of the two reward terms, as shown in line 11-14 of Algorithm 1, allowing the robot to fully explore and learn from both the environment and the expert behaviors.

---

**Algorithm 1** Adaptive Sub-task Skill Learning.
Key differences to T-STAR [15] in gray.

---

1: **Require:** expert demonstrations $\mathbb{D}_1^E, \ldots, \mathbb{D}_K^E$, sub-task MDPs $\mathcal{M}_1, \ldots, \mathcal{M}_K$
2: Initialize sub-task policies $\pi_\theta^1, \ldots, \pi_\theta^K$, least-squares GAIL discriminator $f_\phi^1, \ldots, f_\phi^K$.
3: Initialize imitation progress recognizer $\Phi$ with $\mathbb{D}^E$, balance discount factor $\lambda_{\text{RL}} \leftarrow \alpha$, $\lambda_{\text{IL}} \leftarrow 1 - \alpha$.
4: **for** each sub-task $i = 1, \ldots, K$ **do**
5:   **for** episode $= 1, 2, \ldots, N$ **do**
6:     Rollout trajectories $\tau = (s_1, a_1, r_1^{\text{Env}}, \ldots, s_T)$ with $\pi_\theta^i$
7:     // WEIGHTED REWARD FUNCTION
8:     Compute balanced reward $\{r_1, \ldots, r_{T-1}\} \leftarrow \lambda_{\text{RL}} r^{\text{Env}} + \lambda_{\text{IL}} r^{\text{Pred}}$
9:     Update $f_\phi^i$ with $\tau$ and $\tau^E \sim \mathbb{D}_i^E$ using Eq. 3
10:     Update $\pi_\theta^i$ with the rewarded trajectories $\{s_1, a_1, r_1, \ldots, s_T\}$
11:     // ADAPTIVE EQUILIBRIUM SCHEDULING REGULARIZATION
12:     Update imitation progress recognizer $\Phi$ with $\tau$ and $\tau^E \sim \mathbb{D}_i^E$
13:     Query $\Phi$ about the current imitation progress $p$
14:     Update balance discount factor $\lambda_{\text{RL}}, \lambda_{\text{IL}} \leftarrow \varphi_\lambda(p)$
15:   **end for**
16: **end for**

---

### A.2  Bi-directional Adversarial Learning

As shown in Algorithm 2, the innovation of the bi-directional adversarial learning mechanism consists of two parts: 1) We propose a bi-directional regularization which is trained by two balanced bi-directional constraints to better chain sequential skills, as shown in line 16-17 of Algorithm 2. 2) We also employ the adaptive sub-skill learning scheme during the bi-directional adversarial learning process in order to ensure inter-skill alignment while enabling the sub-task skills to be adaptively adjusted to task changes during fine-tuning as well, as shown in line 10-12 of Algorithm 2.

## B  More Details on AES Regularization

Automatic Discount Scheduling (ADS) [9] is a mechanism for allocating more appropriate reward signals in s Imitation Learning from Observation (ILfO), based on the concept of Optimal Transport [51] and further introducing the characteristic of process dependency across tasks. Based on this, ADS focuses on adjusting the discount factor during reinforcement learning training in ILfO. Following the mechanism in ADS, our AES also employs an imitation progress recognizer $\Phi$ to monitor the extent to which the agent has assimilated the expert's behaviors. The main idea is to assess the closeness of the pair of trajectories by evaluating the agent-collected trajectory $\tau = (s_0, \ldots, s_T)$ and the expert trajectory $\tau^E = (s_0^E, \ldots, s_T^E)$ through a monotonic state-by-state alignment.

**Algorithm 2** Bi-directional Adversarial Learning

Key differences to T-STAR [15] in gray.

1: **Require:** expert demonstrations $\mathbb{D}_1^E, \ldots, \mathbb{D}_K^E$, sub-task MDPs $\mathcal{M}_1, \ldots, \mathcal{M}_K$, pre-trained sub-task policies $\pi_\theta^1, \ldots, \pi_\theta^K$, pre-trained GAIL discriminator $f_\phi^1, \ldots, f_\phi^K$.
2: Initialize bi-directional discriminator $\zeta_\omega^1, \ldots, \zeta_\omega^K$, imitation identifier $\Phi$ with $\mathbb{D}^E$, balance discount factor $\lambda_{\text{RL}} \leftarrow \alpha, \lambda_{\text{IL}} \leftarrow 1 - \alpha$.
3: Initialize initial state buffers $\mathcal{B}_I^1, \ldots, \mathcal{B}_I^K$, and terminal state buffers $\mathcal{B}_\beta^1, \ldots, \mathcal{B}_\beta^K$.
4: **for** iteration $m = 0, 1, \ldots, M$ **do**
5:    **for** each sub-task $i = 1, \ldots, K$ **do**
6:       Sample $s_0$ from environment or $\mathcal{B}_\beta^{i-1}$
7:       Rollout trajectories $\tau = (s_1, a_1, r_1, \ldots, s_T)$ with pre-trained $\pi_\theta^i$
8:       **if** $\tau$ is successful **then**
9:          $\mathcal{B}_I^i \leftarrow \mathcal{B}_I^i \cup s_1, \mathcal{B}_\beta^i \leftarrow \mathcal{B}_\beta^i \cup s_T$
10:          // ADAPTIVE EQUILIBRIUM SCHEDULING
11:          Update imitation identifier $\Phi$ with $\tau$
12:          Query $\Phi$ about the current imitation progress $p$
13:       **end if**
14:       Update balance discount factor $\lambda_{\text{RL}}, \lambda_{\text{IL}} \leftarrow \varphi_\lambda(p)$
15:       Fine-tune $f_\phi^i$ with $\tau$ and $\tau^E \sim \mathbb{D}_i^E$
16:       // TRAIN BI-DIRECTIONAL DISCRIMINATOR
17:       Update $\zeta_\omega^i$ with $s_\beta \sim \mathcal{B}_\beta^{i-1}$ and $s_I \sim \mathcal{B}_I^i$ with $\mathcal{L}_i(\omega) = \frac{1}{2}\mathcal{C}_1 + \frac{1}{2}\mathcal{C}_2$
18:       // FINE-TUNE WITH DUAL REGULARIZATION
19:       Update $\pi_\theta^i$ with $r_i(s_t, a_t, s_{t+1}; \phi, \omega)$ using Eq. 7
20:    **end for**
21: **end for**

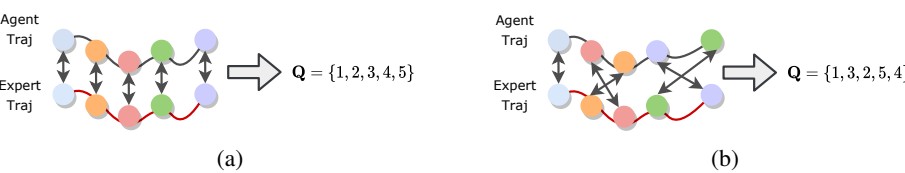

Figure 7: Visualization of the construction of the sequence $\mathbf{Q}$. To be more intuitive, we directly represent the minimum cosine similarity with double arrows.

To be specific, $\Phi$ receives the agent's collected trajectories $\tau$ (line 12 in Algorithm 1) and infers the agent's current imitation progress $p$, $p \in [0, T)$ (line 13 in Algorithm 1). The construction of $\Phi$, with reference to ADS, first requires the construction of a sequence $\mathbf{Q}(q_1, \ldots, q_T)$, where $q_i = \arg\min_j c(s_i, s_j^E)$ is the index of the nearest neighbor of $s_i$ in $\tau^E$, $c$ is the cosine similarity. As shown in Fig. 7, If $\tau$ and $\tau^E$ are exactly the same, then $\mathbf{Q}$ becomes a strictly increasing sequence (Fig 7(a)). On the contrary, if $\tau$ and $\tau^E$ characterize some different behaviors, there are some unordered sequences in $\mathbf{Q}$ (Fig 7(b)).

After constructing $\mathbf{Q}$, the progress alignment between $\tau$ and $\tau^E$ is measured as the length of the longest increasing subsequence (LIS) in $\mathbf{Q}$, denoted as $LIS(\tau, \tau^E)$. For instance, if $\mathbf{Q} = \{1, 3, 2, 5, 4\}$ as in Fig 7(b), then its LIS can be $\{1, 3, 5\}$, $\{1, 2, 5\}$, $\{1, 3, 4\}$ or $\{1, 2, 4\}$. The LIS measurement concentrates on the consistency of the macroscopic trends in these trajectories, thereby preventing overfitting to the microscopic features in the observation [9].

Further, if the following inequality Eq. 8 holds, this indicates that at this time step, the agent's imitation of the expert's action is equivalent to the level of the expert's performance, then the agent's imitation progress $p$ will increase by 1:

$$\max_{\acute{\tau}^E \in \mathbb{D}^E} LIS(\tau_{1:p+1}, \acute{\tau}_{1:p+1}^E) \geq \rho \times \min_{\acute{\tau}^E, \grave{\tau}^E \in \mathbb{D}^E} LIS(\acute{\tau}_{1:p+1}^E, \grave{\tau}_{1:p+1}^E), \tag{8}$$

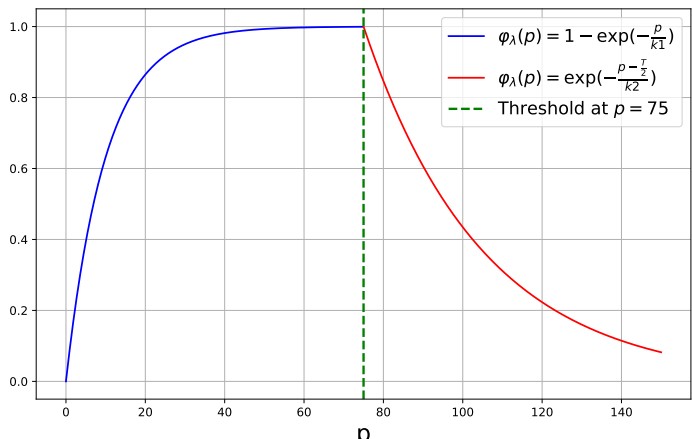

Figure 8: Visualization of the mapping function $\varphi_\lambda(p)$. In this example, we assume that $T = 150$.

where $\acute{\tau}^E \neq \grave{\tau}^E$, the subscript $1 : p + 1$ denotes the first $p + 1$ steps of the extracted trajectory, and $\rho \in [0, 1]$ controls the stringency of the imitation progress monitoring.

After obtaining the current imitation progress $p$ of the agent, AES then adopts a mapping function $\varphi_\lambda(p)$ to schedule the two new balance discount factors $\lambda_{\text{RL}}$ and $\lambda_{\text{IL}}$. Straightforward idea of setting $\varphi_\lambda(p)$ is that **If $p$ reaches a certain threshold, i.e., the agent is able to imitate the expert's behavior well, then the more the agent tends to imitate the expert's behavior in subsequent training, and vice versa.** Therefore, we set the threshold as $\frac{T}{2}$. If $p \in [0, \frac{T}{2})$, we propose $\varphi_\lambda(p) = 1 - e^{\left(-\frac{p}{k1}\right)}$; if $p \in [\frac{T}{2}, T)$, we propose $\varphi_\lambda(p) = e^{\left(-\frac{p-\frac{T}{2}}{k2}\right)}$, where $k1$ and $k2$ are used to flatten the curve of the mapping function. The mapping function shown in Fig. 8, where $T = 150$. In our experiments, we use different flatten factors for the two stages, where $k1 = 10$ and $k2 = 30$.

Then $\lambda_{\text{RL}}$ and $\lambda_{\text{IL}}$ are scheduled to be :

$$
\begin{cases}
\lambda_{\text{RL}} = \alpha^{1-e^{\left(-\frac{p}{k1}\right)}}, \lambda_{\text{IL}} = 1 - \alpha^{1-e^{\left(-\frac{p}{k1}\right)}}. & \text{if } p \in [0, \frac{T}{2}) \\
\lambda_{\text{IL}} = \alpha^{e^{\left(-\frac{p-\frac{T}{2}}{k2}\right)}}, \lambda_{\text{RL}} = 1 - \alpha^{e^{\left(-\frac{p-\frac{T}{2}}{k2}\right)}}. & \text{if } p \in [\frac{T}{2}, T)
\end{cases}
\tag{9}
$$

As shown by the trend of function $\alpha^{\varphi_\lambda(p)}$ in Fig 9, when $p \in [0, \frac{T}{2})$, the scale of $\lambda_{\text{RL}}$: $\alpha^{1-e^{\left(-\frac{p}{k1}\right)}}$ is scheduled to be larger than $\lambda_{\text{IL}}$: $1 - \alpha^{1-e^{\left(-\frac{p}{k1}\right)}}$, but this gap gets smaller and smaller as $p$ gets larger. When $p \in [\frac{T}{2}, T)$, the scale of $\lambda_{\text{IL}}$: $\alpha^{e^{\left(-\frac{p-\frac{T}{2}}{k2}\right)}}$ is scheduled to be larger than $\lambda_{\text{RL}}$: $1 - \alpha^{e^{\left(-\frac{p-\frac{T}{2}}{k2}\right)}}$, while the scale of $\lambda_{\text{IL}}$ increases as the agent imitates better.

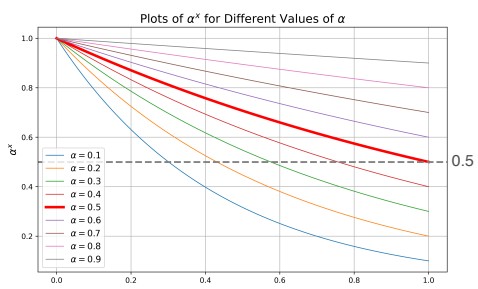

Figure 9: $\alpha^{\varphi_\lambda(p)}$ based on the variation of different $\alpha$ sizes in $\varphi_\lambda(p) \in [0, 1]$. We use $\alpha = 0.5$ as the base in our experiments.

**Thus, if $p$ is larger and reaches a threshold step, i.e., the agent is able to imitate the expert's behavior well, then the more the agent tends to imitate the expert's behavior in subsequent training, and vice versa.** The entire process is adaptively scheduled based on $\Phi$ periodic monitoring of the agent's imitation process. Consequently, the RL and IL components of

sub-task skill learning can be adaptively scheduled and regularized through AES, effectively enhancing *intra-skill dependencies* between sequential actions.

## C    Sub-task Skills

In our simulation experiments, we use sequences of sub-tasks defined internally by the environment [44, 45] as task decomposition sub-tasks. Here we list these sequential skills to emphasize the difficulty of long-horizon tasks. Each skill takes a 3D position as the input $g_*$.

**IKEA Furniture Assembly:**

**Chair_agne (2 sub-task skills):** Assemble stool leg 0 to target position $g_*^0 \to$ Assemble stool leg 1 to target position $g_*^1$

**Chair_bernhard (2 sub-task skills):** Assemble support leg 0 to target position $g_*^0 \to$ Assemble support leg 1 to target position $g_*^1$

**Table_dockstra (3 sub-task skills):** Assemble table leg 0 to target position $g_*^0 \to$ Assemble table leg 1 to target position $g_*^1 \to$ Assemble table top to target position $g_*^3$

**Chair_ingolf (4 sub-task skills):** Assemble chair support 0 to target position $g_*^0 \to$ Assemble chair support 1 to target position $g_*^1 \to$ Assemble front leg 0 to target position $g_*^3 \to$ Assemble front leg 1 to target position $g_*^4$

**Table_lack (4 sub-task skills):** Assemble table leg 0 to target position $g_*^0 \to$ Assemble table leg 1 to target position $g_*^1 \to$ Assemble table leg 2 to target position $g_*^3 \to$ Assemble table leg 3 to target position $g_*^4$

**Toy_table (4 sub-task skills):** Assemble table leg 0 insert to target position $g_*^0 \to$ Assemble table leg 1 insert to target position $g_*^1 \to$ Assemble table leg 2 insert to target position $g_*^3 \to$ Assemble table leg 3 insert to target position $g_*^4$

**Kitchen Organization:**

**Kitchen (4 sub-task skills):** Turn on the microwave to target position $g_*^0 \to$ Move the kettle to target position $g_*^1 \to$ Turn on the stove (rotate the stove button to target position $g_*^2$) $\to$ Turn on the light (rotate the light button to target position $g_*^3$)

**Extended Kitchen (5 sub-task skills):** Turn on the microwave to target position $g_*^0 \to$ Turn on the stove (rotate the stove button to target position $g_*^1$) $\to$ Turn on the light (rotate the light button to target position $g_*^2$) $\to$ Slide the cabinet to the right target position $g_*^3 \to$ Open the cabinet to target position $g_*^4$

## D    More Quantitative Results

We present the training curves with different skill learning methods for sub-task skills in *chair_ingolf* task, and we further present the evaluation performance of the pre-trained skills with different methods across sub-tasks in the other 6 long-horizon simulation tasks. Also, we test the algorithms trained from scratch in the presence of perturbations to further illustrate the importance of the execution of sub-tasks on long-horizon tasks.

Additionally, the main paper does not delve into the loss function $\mathcal{L}_i(\omega)$ concerning the different scales of the bi-directional constraints in bi-directional adversarial training. Therefore, we conduct further ablation experiments to examine the impact of different scales of the two constraints in the bi-directional discriminator.

### D.1    Sub-task Skill Learning Performance

### D.1.1    Training performance

Fig. 10 shows the sub-task skill training curves in IKEA furniture assembly tasks. All methods are trained in each sub-task with 5 random seeds, 15M environment steps. As can be seen, the sub-task skill training based on PPO (learning only from environmental feedback), GAIL (learning only from expert demonstrations) and Fixed-RL-IL (learning from a fixed scale of environmental feedback

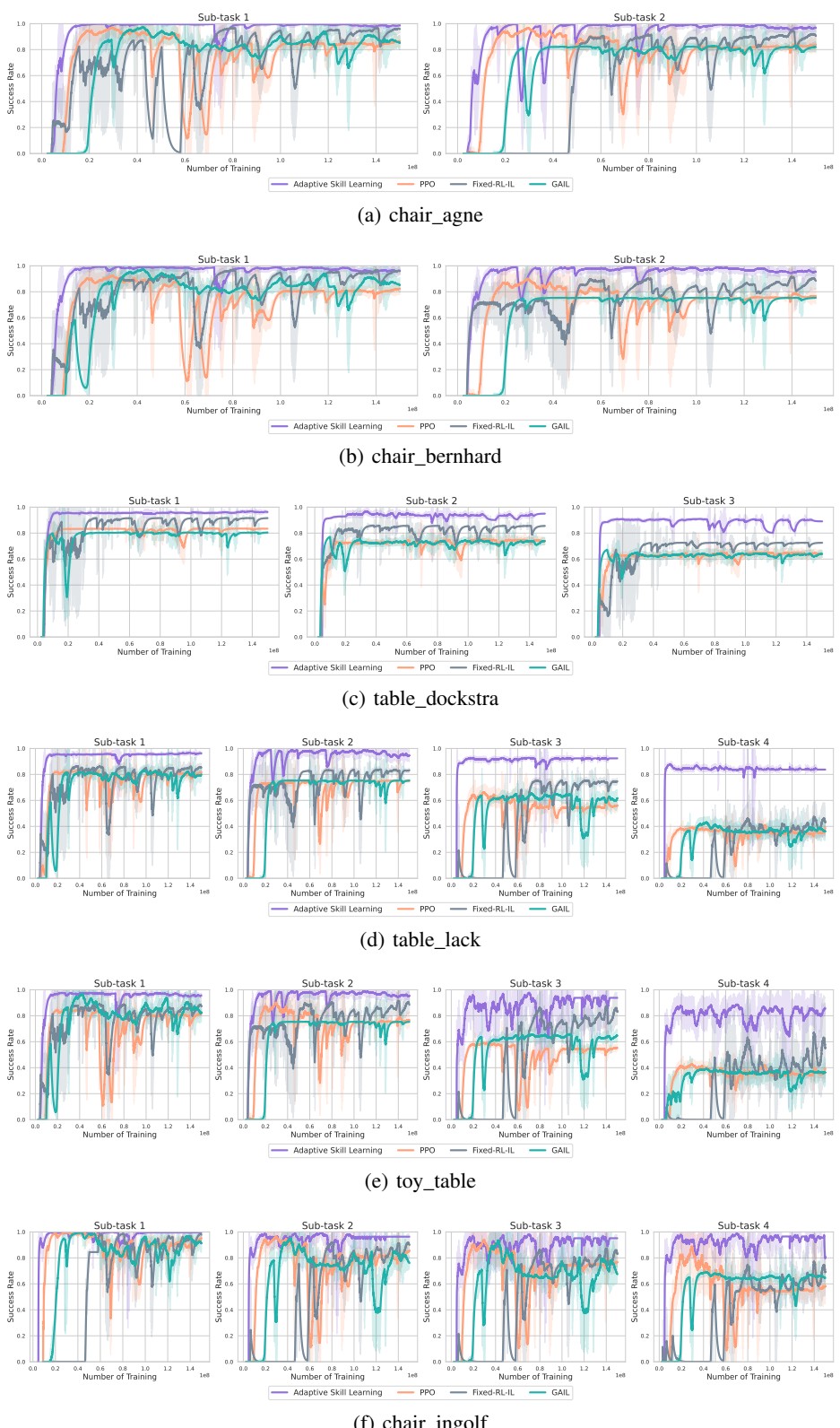

Figure 10: Training curves for sub-task skills in IKEA furniture assembly tasks. The y-axis represents the success rate of the sub-task.

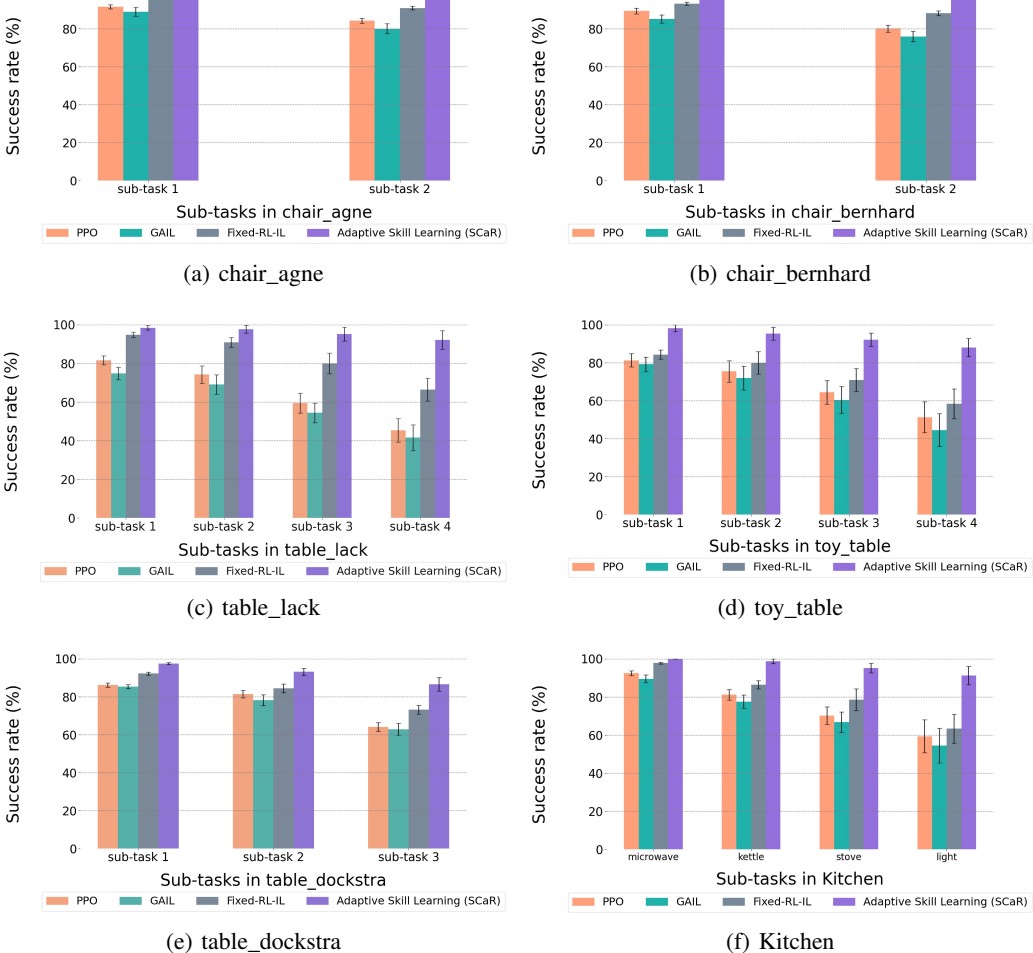

Figure 11: Evaluation Performance Comparison of Sub-task Skill Learning.

and expert demonstration) cannot maintain stability and exhibits significant training performance degradation as the sub-task stage increases. In contrast, the sub-task skill training process using our proposed adaptive sub-skill learning scheme has always been relatively stable and better performing.

### D.1.2 More evaluation performance

As shown in Fig. 11, in *chair_agne*, *chair_bernhard*, *table_lacktoy_table*, *table_dockstra*, and Kitchen tasks, even with the increase of objects in the environment - and the increase of unpredictable perturbations - our proposed adaptive skill learning learns better sub-task skills. In contrast, the PPO, GAIL, and Fixed-RL-IL baselines fail to maintain well-learning sub-task skills.

These results further corroborate that our proposed AES regularization can reinforce *inter-step dependencies* to the sequential actions within each sub-task skill, and thus pre-train better sub-task skills for long-horizon tasks.

### D.2 Robustness to Perturbations

We test the algorithms trained from scratch in the presence of perturbations. As shown in Table 3, algorithms trained from scratch fail to successfully complete the task when environment perturbations occur during execution. This further illustrates the importance of dividing sub-tasks for multi-stage execution on long-horizon manipulation tasks that are contact-rich and subject to unanticipated

Table 3: Success rates of completing the two sub-tasks *chair_bernhard* and four sub-tasks *chair_ingolf* in stationary and perturbed environments.

| | *chair_bernhard* | | *chair_ingolf* | |
|---|---|---|---|---|
| Method | **No Perturb** | **Perturb** | **No Perturb** | **Perturb** |
| **PPO (Scratch RL)** | $0.42 \pm 0.12$ | $0.01 \pm 0.00$ | $0.14 \pm 0.03$ | $0.00 \pm 0.00$ |
| **GAIL (Scratch IL)** | $0.23 \pm 0.02$ | $0.00 \pm 0.00$ | $0.00 \pm 0.00$ | $0.00 \pm 0.00$ |
| **Fixed-RL-IL** | $0.53 \pm 0.07$ | $0.05 \pm 0.00$ | $0.22 \pm 0.00$ | $0.00 \pm 0.00$ |
| **SkiMo** | $0.62 \pm 0.05$ | $0.10 \pm 0.00$ | $0.47 \pm 0.03$ | $0.00 \pm 0.00$ |
| **Policy Sequencing** | $0.82 \pm 0.09$ | $0.51 \pm 0.04$ | $0.77 \pm 0.12$ | $0.50 \pm 0.10$ |
| **T-STAR** | $0.90 \pm 0.04$ | $0.60 \pm 0.08$ | $0.89 \pm 0.04$ | $0.59 \pm 0.04$ |
| **SCaR w/o Bi** | $0.92 \pm 0.02$ | $0.65 \pm 0.11$ | $0.91 \pm 0.01$ | $0.63 \pm 0.05$ |
| **SCaR w/o AES** | $0.94 \pm 0.03$ | $0.74 \pm 0.09$ | $0.93 \pm 0.02$ | $0.71 \pm 0.07$ |
| **SCaR (Ours)** | $\mathbf{0.96} \pm \mathbf{0.04}$ | $\mathbf{0.85} \pm \mathbf{0.11}$ | $\mathbf{0.95} \pm \mathbf{0.03}$ | $\mathbf{0.80} \pm \mathbf{0.13}$ |

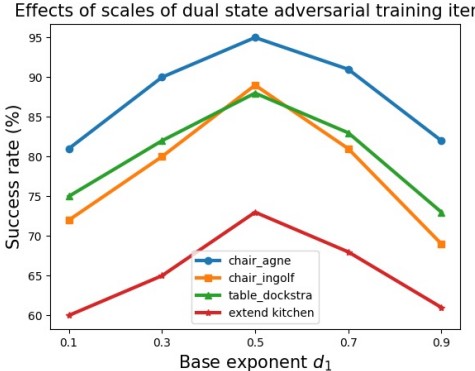

Figure 12: Impact on skill chaining performance of different scales of bi-directional constraints in SCaR.

perturbations. It also supports the significance of our work on long-horizon robotic manipulation tasks.

### D.3 Further Ablation

We set the loss function for the bi-directional discriminator in the main paper as $\mathcal{L}_i(\omega) = \frac{1}{2}\mathcal{C}_1 + \frac{1}{2}\mathcal{C}_2$., where the bi-directional constraints $\mathcal{C}_1, \mathcal{C}_2$ are defined as:

$$\textbf{next initial} \rightarrow \textbf{previous terminal:} \quad \mathcal{C}_1 = \mathbb{E}_{s_{\mathcal{I}} \sim \mathcal{I}_i}[\zeta_\omega^i(s_{\mathcal{I}}) - 1]^2 + \mathbb{E}_{s_T \sim \beta_{i-1}}[\zeta_\omega^i(s_T)]^2$$
$$\textbf{previous terminal} \rightarrow \textbf{next initial:} \quad \mathcal{C}_2 = \mathbb{E}_{s_T \sim \beta_i}[\zeta_\omega^i(s_T) - 1]^2 + \mathbb{E}_{s_{\mathcal{I}} \sim \mathcal{I}_{i+1}}[\zeta_\omega^i(s_{\mathcal{I}})]^2 \tag{10}$$

The first constraint $\mathcal{C}_1$ trains the policy to have the initial states approach the terminal states of the previous policy, while the second constraint $\mathcal{C}_2$ trains the policy to have the terminal states close to the initial states of the next policy. In the experiments, these two constraints have the same scale in the training process of the bi-directional discriminator.

We wonder whether different scales of these two terms would lead to different performances, and for this reason, we conduct further parametric ablation experiments to explore this. Specifically, we define the scale parameter of the first term $\mathcal{C}_1$ as $d_1$, and the second term $\mathcal{C}_2$ as $d_2 = 1 - d_1$, and set **0.1, 0.3, 0.5, 0.7, 0.9** for $d_1$ respectively for comparison experiments. We test the effect of different scales of bi-directional adversarial training items $d_1$ and $d_2$ on the success rate of SCaR in each of the four tasks: *chair_agne*, *chair_ingolf*, *table_dockstra*, and extend kitchen. As shown in Fig. 12, the experimental result is also in line with our intuition that when the ratio of the two terms **initial** → **previous terminal** and **terminal** → **next initial** is the same, the performance is the best

among the four tasks, whereas when the more imbalanced the scale of the two terms is, the worse the performance is.

This ablation result further demonstrate our statement in Sec. 4.3 in the main paper: **The purpose of the bi-directional discriminator is to establish a balanced mapping relationship between the initial states and terminal states to ensure the coherence and stability of the policy.** If the constraint in one direction (e.g., from initial states to terminal states) is stronger than the constraint in the other direction (e.g., from terminal states to initial states), the information transmission becomes asymmetric. This asymmetry results in better training in one direction and insufficient training in the other, thereby affecting overall performance.

### D.4  Impact of Different Sub-task Divisions

To explore the impact of different sub-task divisions, we conduct more experimental validation using the chair_ingolf task. The original sub-tasks in this task are divided as follows: "Assemble chair support 0 to target position" → "Assemble chair support 1 to target position" → "Assemble front leg 0 to target position" → "Assemble front leg 1 to target position". We have re-divided the sub-tasks into two alternative settings:

1. "Assemble chair support 0 and chair support 1 to target positions" → "Assemble front leg 0 to target position" → "Assemble front leg 1 to target position".

2. "Assemble chair support 0 to target position" → "Assemble chair support 1 to target position" → "Assemble front leg 0 and leg 1 to target positions".

Table 4: The impact of different sub-task divisions on SCaR performance.

|  | chair_ingolf (setup 1) | chair_ingolf (setup 2) |
|---|---|---|
| SCaR | 0.68 | 0.74 |

It is worth noting that, since the re-division of the sub-tasks results in only three sub-tasks, we set 90% as the success metric for all three sub-tasks being successfully executed. As seen in Table 4, compared to SCaR's success rate of about 95% with the original four sub-task divisions, the success rate for completing the first sub-task and then executing the remaining two sub-tasks is significantly reduced. This decrease is due to the increased difficulty of the first sub-task in setup 1 (which requires assembling both chair support) and the last sub-task in setup 2. These changes result in a lower overall success rate for the task. This result suggests that a reasonable division of sub-tasks in long-horizon tasks is crucial for the success rate of overall task completion.

### D.5  Impact of the Number of Sub-tasks

To explore the performance of skill-chaining methods as the number of sub-tasks increases, we add a sub-task to the **Extended Kitchen** task to evaluate SCaR's performance in manipulation tasks with longer horizons, involving 6 sub-tasks. The modified task, **Longer Extended Kitchen** includes: 1. Turn on the microwave; 2. Turn on the stove; 3. Turn on the light; 4. Slide the cabinet to the right target position; 5. Open the cabinet to the target position; **6. Move the kettle to the target position**.

Table 5: Performance Comparison on Longer Extended Kitchen Task.

| Method | Longer Extended Kitchen Task |
|---|---|
| T-STAR | 0.33 |
| SCaR | 0.61 |

As shown in Table 5, the addition of an extra sub-task increases the complexity and difficulty of skill chaining in long-horizon tasks. Nonetheless, SCaR achieves a significantly higher overall task execution success rate, surpassing T-STAR by 28%. Although there is still ample room for improvement, we believe our approach establishes a strong baseline for future research on skill-chaining methods for long-horizon manipulation tasks.

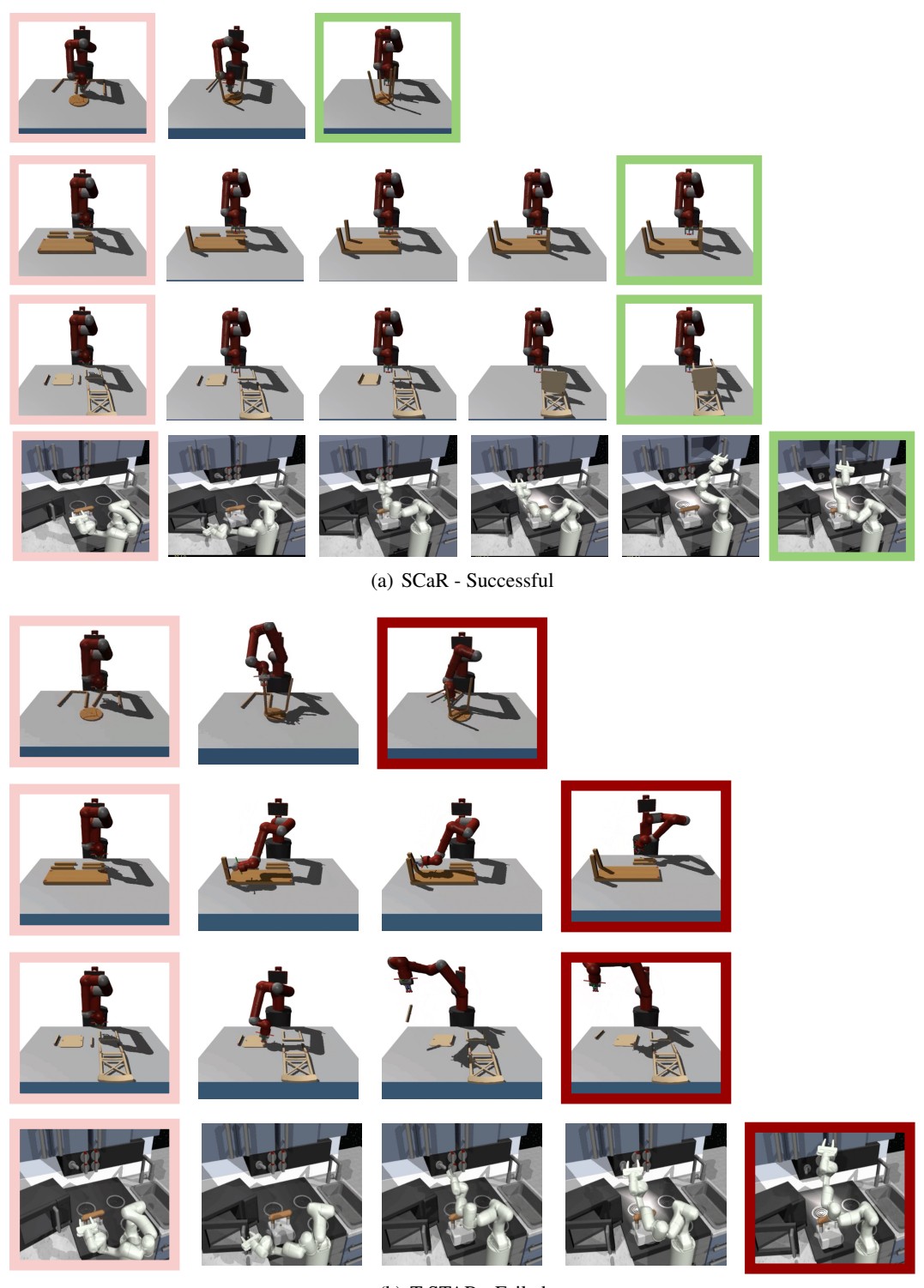

(a) SCaR - Successful

(b) T-STAR - Failed

Figure 13: Qualitative results of successful skill chaining performance with SCaR and failed skill chaining performance with T-STAR. More qualitative results can be found on our project website https://sites.google.com/view/scar8297.

# E   More Qualitative Results

Fig 13 shows the qualitative comparison of skill chaining methods. Their animated versions can be found on our project website.

# F   Real-world Validation via Sim-to-Real Transfer

Table 6: Skill chaining performance of real-world long-horizon robotic manipulation tasks.

| Method | Success rate |
|--------|--------------|
| T-STAR | 70% (2 sub-tasks) / 50% (3 sub-tasks) |
| SCaR | **90%** (2 sub-tasks) / **70%** (3 sub-tasks) |

**Real-robot Experiment Setup**    We also evaluate the skill chaining performance of real-robot for solving simple yet intuitive real-world long-horizon manipulation. We set up two types of desktop-level long-horizon manipulation tasks. The robotic arm needs to pick-and-place 2 and 3 blue squares in sequence.

We built the corresponding task environment using the gazebo simulation that accompanies the K1 robot[5], and collect 50 demonstrations of grasping skills for each square for training. With camera calibration, we deploy agents trained under simulation in a real robot desktop task to solve 2-square as well as 3-square pick-and-place tasks without the need for adaptation processes. We conduct experiments with the Sagittarius K1 and use MoveIt2 library based on ROS 2 framework for controlling the arm. We use RGB observations from RealSense D435i camera on the wrist of the robotic arm.

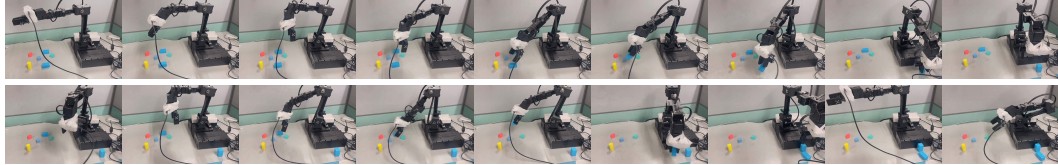

Figure 14: Visualization of the successful skill chaining in the 2-blue-square pick-and-place tasks using SCaR.

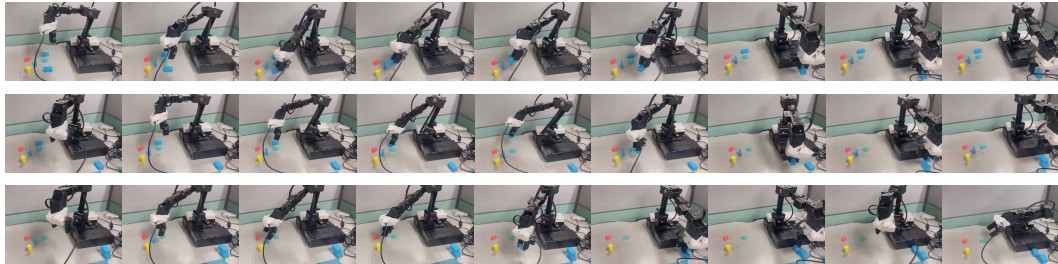

Figure 15: Visualization of the successful skill chaining in the 3-blue-square pick-and-place tasks using SCaR.

**Results**    For evaluation, we measure the success rate across 10 randomized square positions for each task. As shown in Table. 6, SCaR can solve the two long-horizon tasks and outperforms T-STAR baseline. Fig. 14 and Fig. 15 show the qualitative results of successful skill chaining in the 2 and 3-blue-square pick-and-place tasks using SCaR.

---

[5]https://github.com/NXROBO/sagittarius_ws

Figure 16: **IKEA Furniture Assembly Environment for Long-Horizon Complex Manipulation Tasks.**

# G    Environment Details

## G.1    IKEA Furniture Assembly

We choose six tasks, *chair_agne*, *chair_bernhard*, *chair_ingolf*, *table_lack*, *toy_table*, and *table_dockstra* from the IKEA furniture assembly environment[6] [44] as the focal points of our experiments, as shown in Fig. 17. Our chosen robotic platform is the 7-DoF Rethink Sawyer robot, and we control it using joint velocity commands.

**Observation Space**    The observation space comprises three key components: robot observations (29 dimensions), object observations (35 dimensions), and task phase information (8 dimensions). Robot observations encompass robot joint angles (7 dimensions), joint velocities (7 dimensions), gripper state (2 dimensions), gripper position (3 dimensions), gripper quaternion (4 dimensions), gripper velocity (3 dimensions), and gripper angular velocity (3 dimensions). Object observations include the positions (3 dimensions) and quaternions (4 dimensions) of all five furniture pieces in the scene. Task information, an 8-dimensional one-hot encoding, represents the current phase, including actions like reaching, grasping, lifting, moving, and aligning.

**Action space**    The action space includes arm movement, gripper control, and the connect action, which can vary based on different control modes: 6D end-effector space control using inverse kinematics, joint velocity control, and joint torque control.

In the context of reinforcement learning (RL), we utilize a heavily shaped multi-phase dense reward obtained from the IKEA Furniture Assembly Environment [44].

**Environmental Reward Function**    The IKEA furniture assembly environmental reward function is a multi-phase reward defined with respect to a pair of furniture parts to attach (e.g., a table leg and a table top) and the corresponding manually annotated way-points, such as a target gripping point $g$ for each part. The reward function for a pair of furniture parts consists of eight different phases as follows:

- **Initial phase:** The robot has to reconfigure its arm pose to an appropriate pose $\mathbf{p}_{\text{init}}$ for grasping a new furniture part. The reward is proportional to the negative distance between the end-effector $\mathbf{p}_{\text{eff}}$ and $\mathbf{p}_{\text{init}}$.

- **Reach phase:** The robot reaches above a target furniture part. The reward is proportional to the negative distance between the end-effector $\mathbf{p}_{\text{eff}}$ and a point $\mathbf{p}_{\text{reach}}$ 5 cm above the gripping point $g$.

- **Lower phase:** The gripper is lowered onto the target part. The phase reward is proportional to the negative distance between $\mathbf{p}_{\text{eff}}$ and the target gripping points.

- **Grasp phase:** The robot learns to grasp the target part. The reward is given if the gripper contacts the part, and is proportional to the force exerted by the grippers.

- **Lift phase:** The robot lifts the gripped part up to $\mathbf{p}_{\text{lift}}$. The reward is proportional to the negative distance between the gripped part $\mathbf{p}_{\text{part}}$ and the target point $\mathbf{p}_{\text{lift}}$.

- **Align phase:** The robot roughly rotates the gripped part before moving it. The reward is proportional to the cosine similarity between up vectors $\mathbf{u}_A$, $\mathbf{u}_B$ and forward vectors $\mathbf{f}_A$, $\mathbf{f}_B$ of the two connectors.

---

[6]https://github.com/clvrai/furniture



**Kitchen**            **Extended Kitchen**

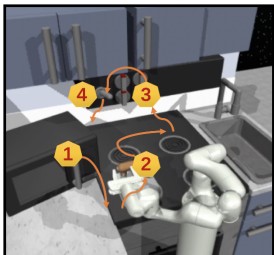 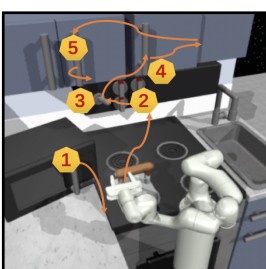



Figure 17: **Kitchen Organization Environment for Long-Horizon Complex Manipulation Tasks.**

- **Move phase:** The robot moves and aligns the gripped part to another part. The reward is proportional to the negative distance between the connector of the gripped part and a point $\mathbf{p}_{\text{move\_to}}$ 5 cm above the connector of another part, and the cosine similarity between two connector up vectors, $\mathbf{u}_A$ and $\mathbf{u}_B$, and forward vectors $\mathbf{f}_A$ and $\mathbf{f}_B$. Note that all connectors are labeled with aligned up vectors and forward vectors.

- **Fine-grained move phase:** The robot must finely align two connectors until attached. The same reward is used as the **move phase** with a higher coefficient, making the reward more sensitive to small changes. In addition, when the part is connectable, a reward is provided based on the activation of the connect action $a[\text{connect}]$.

Upon completion of each phase, completion rewards are given to encourage the agent to move on to the next phase. In addition to stage-based rewards, control penalties, stabilizing wrist pose rewards, and grasping rewards (i.e., opening the grasping hand only during the initial, arrival, and lower stages) are provided throughout the process. If the robot releases the grasped object, the phase ends early and a negative reward is provided. Phase completion depends on the robot and part configurations satisfying distance and angle constraints with respect to the goal configuration. After all stages are completed, the stage resets to the initial stage. This process repeats until all parts are connected.

**Demonstration Collection**     For imitation learning (IL), we gathered 200 demonstrations for each furniture part assembly using a programmatic assembly policy. Each demonstration for single-part assembly typically spans 150 steps, reflecting the overall task's inherently long-horizon nature.

**Sub-tasks**     In our experiments, we define a sub-task as the process of assembling one part to another. Thus, the *chair_agne* and *chair_bernhard* tasks have two distinct sub-tasks, *table_dockstra* has three distinct sub-tasks, and *chair_ingolf*, *table_lack*, and *toy_table* have four distinct sub-tasks. These sub-tasks are trained independently, with their initial state sampled from the environment and random noise introduced in the [-2cm, 2cm] and [-3°, 3°] ranges of the (x, y) plane. Importantly, the decomposition of the sub-tasks is pre-determined, which means that the environment is initialized for each sub-task, and the agent receives a notification when a sub-task is successfully completed. Once the two components are firmly connected, the corresponding sub-task is considered completed and the robotic arm is guided back to its initial pose, i.e., at the center of the workspace.

**Assembly Difficulty**     The difficulty of modeling furniture depends largely on the shape of the furniture. For example, the *toy_table* task with cylindrical legs is more difficult to grasp, whereas the *table_lack* task with rectangular legs is easier to grasp. Chairs are generally more difficult to assemble because of their irregular shape (e.g., seat and back). This is the reason why the success rates of the *toy_table* and *chair_ingolf* tasks are lower than the success rates of *table_lack*.

### G.2 Kitchen Organization

We use the Franka Kitchen tasks in D4RL [45] and refer to the experimental setup in SkiMo [46] for the sub-task extensions. Including the following two tasks: **Kitchen task** and **Extended Kitchen task**, as shown in Fig. 17.

**Kitchen**    The 7-DoF Franka Emika Panda robot arm is tasked with performing four sequential sub-tasks: *Turn on the microwave - Move the kettle - Turn on the stove - Turn on the lights*.

**Extended Kitchen**    The environment and task-agnostic data used in this experiment are consistent with those employed in the **Kitchen** scenario. However, we introduce a different set of sub-tasks for this experiment, namely: *Turn on the microwave - Turn on the stove - Turn on the lights - Slide the cabinets to the right - Open the cabinets*, as depicted in Fig. 17 (right). It's worth noting that this sequence of tasks is not aligned with the sub-task transition probabilities observed in the task-agnostic data, posing a challenge for exploration based on prior data.

**Observation Space**    The agent operates within a 30-dimensional observation space, which includes an 11-dimensional robot proprioceptive state and 19-dimensional object states. This modified observation space removes a constant 30-dimensional goal state found in the original environment.

**Action Space**    The agent's action space consists of 9 dimensions, encompassing 7-dimensional joint velocity control and 2-dimensional gripper velocity control.

**Environmental Reward Function**    In terms of the environmental rewards, the agent receives a reward of +1 for each completed sub-task. The total episode length is set to 280 steps, and an episode concludes once all sub-tasks are successfully accomplished. The initial state is initialized with slight noise introduced in each state dimension.

**Demonstration Collection**    For imitation learning, we collect 200 demonstrations per sub-task with reference to the dataset in [52] that obtained through teleoperation. This dataset covers interactions with all seven manipulatable objects within the environment.

## H    Network Architecture

For a fair comparison, our method and the benchmark methods use the same network structure. The policy network and the critic network consist of two layers of 128 and 256 hidden units fully connected with ReLU nonlinear properties, respectively. The output layer of the actor network outputs an action distribution, which consists of the mean and standard deviation of a Gaussian distribution. The critic network outputs only one critic value. The discriminator of GAIL [39] and the bi-directional discriminator of our proposed approach use a two-layer fully connected network with 256 hidden units. The outputs of these discriminators are clipped between [0, 1], following the least-square GAIL proposed by [40].

## I    Training Details

### I.1    Computing Resources

Our method and all baselines were implemented using PyTorch [53]. All experiments were conducted on workstations equipped with Intel(R) Xeon(R) Gold 5218 CPUs and dual NVIDIA GeForce RTX 3080 GPUs. In the SCaR framework, the pre-training phase for each sub-task skill policy, spanning 150M time steps, took approximately 10 hours with dual regularization, compared to about 8 hours without it (Fixed-RL-IL), leading to an added computational overhead of roughly 2 hours. The full testing and evaluation process of skill chaining for a complete long-horizon task required an additional 10 to 15 hours, depending on the task's complexity. For comparison, training the skill dynamics model in SkiMo [46] took approximately 24 hours (100M steps). Training baselines such as PPO [47], GAIL [39], and Fixed-RL-IL took longer, requiring about 48 hours each, as these methods train the entire long-horizon task from scratch, with 450M time steps for each complete task. In our evaluation, we used 5 seeds, each tested over 200 episodes, resulting in an average real-time execution time of about 36-54 seconds per single long-horizon task.

### I.2    Algorithm Implementation Details

We report the hyperparameters used in our experiments in Table 7.

Table 7: Hyperparameters used in our experiments.

| Hyperparameter | Value |
|---|---|
| Rollout Size | 1024 |
| Learning Rate | 0.0003 |
| Learning Rate Decay | Linear decay |
| Mini-batch Size | 128 |
| Discount Factor | 0.99 |
| Entropy Coefficient | 0.003 |
| Reward Scale | 0.05 |
| State Normalization | True |
| Discriminator learning rate | $1e^{-4}$ |
| Sub-task training steps | 150000000 |
| # Workers | 20 |
| # Epochs per Update | 10 |
| Base exponent for balancing $\alpha$ | 0.5 |
| $k_1$ (used to flatten the mapping function during $p \in [0, \frac{T}{2}))$ | 10 |
| $k_2$ (used to flatten the mapping function during $p \in [\frac{T}{2}, T))$ | 30 |
| Weighting factor $\lambda_{\mathrm{Bi}}$ | 10000 |
| $\rho$ (for imitation progress recognizer $\Phi$) | 0.9 |
| Penalty coefficient $\eta^{\mathrm{gp}}$ | 10 |

Table 8: Comparison to prior work and ablated methods.

| Method | Model-based | Skill-based | Scratch training | Joint training |
|---|---|---|---|---|
| PPO [47] and GAIL [39] | ✗ | ✗ | ✓ | ✗ |
| Fixed-RL-IL [40] | ✗ | ✗ | ✓ | ✓ |
| SkiMo [46] | ✓ | ✓ | ✓ | ✓ |
| Policy Sequencing [12] | ✓ | ✓ | ✗ | ✓ |
| T-STAR [15] | ✗ | ✓ | ✗ | ✓ |
| SCaR (Ours) and SCaR w/o Bi and SCaR w/o AES | ✓ | ✓ | ✗ | ✓ |

For the baseline implementations, we use the official code for PPO [47], GAIL [39], Fixed-RL-IL [40], SkiMo [46], Policy Sequencing [12] and T-STAR [15]. The table below (Table 8) compares key components of **SCaR** with model-based, model-free and skill-based baselines and ablated methods, where *joint training* indicates whether or not reinforcement learning combined with imitation learning is used for training.

**PPO [47]**  Any reinforcement learning algorithm can be used for policy optimization, in this paper we choose to use Proximal Policy Optimization (PPO) and use the default hyperparameters of PPO [47].

**GAIL [39]**  In this paper we choose to use Generative Adversarial Imitation Learning (GAIL) [39] as the learning algorithm for imitation learning and use the default hyperparameters of GAIL [39]. We specifically use an agent states $s$ to discriminate agent and expert trajectories, instead of state-action pairs $(s, a)$.

**Fixed-RL-IL [12]**  We adopt the AMP [40] solution combining environmental rewards and least square GAIL with $\lambda_{\mathrm{RL}} = \lambda_{\mathrm{IL}} = 0.5$. For implementation details of least square GAIL training and GAIL rewards, see original paper [40].

**SkiMo [46]**  We use the official implementation of the original paper and use the hyperparameters suggested in the official implementation.

**Policy Sequencing [12]**  We employ the official implementation and the hyperparameters provided by [15].

**T-STAR [15]**    We use the official implementation of the original paper and use the hyperparameters suggested in the official implementation [15].

**SCaR (ours)**    We refer to T-STAR and use $\lambda_{\mathrm{Re}} = 10000$ for bi-directional regularization. We take 50% of the initial state samples from the start environment of each policy, 50% of the terminal state samples at the end, and 50% of the initial state buffer and 50% of the terminal state buffer from the previous skill, respectively.

## J    Limitations and Potential Solutions

While SCaR demonstrates strong capabilities, it does have certain limitations. In the following sections, we outline these limitations and propose potential solutions.

**Limited Task Generalization**    SCaR primarily focuses on predefined, structured environments to validate its mechanism for chaining pre-trained sub-task skills within long-horizon tasks. Consequently, our current study does not address SCaR's adaptability across varied or novel task environments. While we demonstrate SCaR's robustness to unknown perturbations within a single environment (e.g., unexpected forces applied to the robot arm joints in specific sub-tasks), the system's ability to generalize to entirely new or unfamiliar environments remains unexplored. A potential solution is to leverage foundational models to expand SCaR's applicability. In long-horizon tasks where sub-task definitions are unclear or missing, foundational models can use their powerful task-planning capabilities to divide tasks into logical sub-tasks [54, 55, 56].When unexpected sub-task demands or changes in overall task goals arise, these models can re-plan sub-tasks accordingly. Another approach involves human-in-the-loop learning, incorporating human guidance in the training pipeline, such as using human priors for sub-task division [32] or employing methods to manage the subjectivity of human-labeled rewards through preference learning [57].

**Extensive Retraining for New Tasks**    SCaR faces limitations in adapting to diverse long-horizon manipulation tasks or different robotic setups, as it requires extensive retraining of sub-task skills for each new task. This reliance on full retraining restricts SCaR's efficiency and scalability when addressing evolving or variable task requirements, particularly in complex, real-world environments. A potential solution is to integrate online reinforcement learning into SCaR's skill-learning process, allowing it to adapt to task variations, such as different table designs, without full retraining, enabling more efficient adaptation.

**Lack of visual input handling**    SCaR currently lacks the capability to process encoded image and semantic state inputs, limiting its applicability in tasks that rely on visual or semantic information for effective performance. The current setup is largely dependent on environments with direct access to state information, as both simulated environments are based on state representations. While effective in controlled setups, enabling SCaR to learn from image encodings as states would enhance its robustness and applicability in tasks requiring nuanced visual and semantic processing. Given recent advancements in learning multi-view representations [58, 59] and generating 3D models [60, 61], incorporating these improvements and 3D priors into the encoder design could enhance the sample efficiency of SCaR.

## K    Potential negative impacts

Since our method is currently limited to applications in simulated environments and simple desktop-level robot manipulation, it is not expected to have a significant negative impact on society. However, privacy concerns may arise if our method is applied to real-world long time-series tasks with mobility, as imitation learning agents used in applications such as autonomous driving [62] or real-time control [63, 64] require large amounts of data that often contain controversial information. In addition, the imitation learning policy is a challenge because it imitates a specified demonstration that may include bad behavior. If the expert demonstration includes some nefarious behaviors (e.g., training data for a mobile manipulation task includes behaviors that may be violent towards pedestrians), then the policy may have a significant negative impact on the user. To address this issue, future directions should focus on developing agents with safety adaptations in addition to improving performance.

