# OpenReview forum: "SCaR: Refining Skill Chaining for Long-Horizon Robotic Manipulation via Dual Regularization"
_NeurIPS.cc/2024/Conference — NeurIPS 2024 poster_

### Official Review · Reviewer_ysj9 · 2024-07-12

**Soundness:** 2
**Presentation:** 3
**Contribution:** 2
**Rating:** 6
**Confidence:** 4

**Summary:**

This paper focuses on skill chaining for long-horizon robotic manipulation tasks. A dual regularization method is proposed to tackle this problem, including an adaptive sub-task learning scheme that enhance intra-skill dependencies, and a bi-directional adversarial learning mechanism for reinforcing inter-skill dependencies. The proposed framework is evaluated on two long-horizon robotic manipulation simulation benchmarks and real-world robot pick-and-place tasks. Results show that the proposed method outperforms benchmarks on success rate and is more robust to perturbations.

**Strengths:**

Applying dual regularization to skill chaining for long-horizon manipulation tasks is well-motivated by the two common failure cases--failed pre-training of sub-task skills and failed skill chaining due to disturbance.

Extensive experiments in simulation are performed to demonstrate the effectiveness and robustness of the proposed method. Detailed ablations are performed to validate the effectiveness of each component of the approach.

**Weaknesses:**

Real-world evaluation is only done on desktop pick and place task, which is a well-studied task that has many effective and robust solutions, and does not necessarily require intra-skill dependency and inter-skill dependency. The method would have better soundness if it is evaluated on more useful real-world robotic tasks.

The writing of the paper can be improved, especially the figures in the paper are hard to see--the robot execution frames are too small.

**Questions:**

How is the sub-task division of the long-horizon task defined and how will different divisions affect performance?

How well can the method scale to longer-horizon manipulation tasks? How would the method perform when the number of sub-tasks increases?

**Limitations:**

The authors have adequately addressed the limitations.

---

> ### Author Rebuttal · Authors · 2024-08-06
>
> We sincerely thank you for acknowledging the strength of our work, e.g., reasonable idea of dual regularization and extensive experiments. We address the comments and questions below.
>
> **Regarding real-world robotics experiments.**
>
> * Thank you for pointing this out. Due to hardware limitations, we are  currently using simple long-horizon pick-and-place tasks to validate  SCaR's skill-chaining performance in real-world robotics. This is the  same limitation and future direction discussed in the "Limitations and  Future Directions" section of the original paper. We are in the process  of procuring a Franka robotic arm, which we expect will allow us to  validate SCaR on more complex and extensive long-range robot  manipulation tasks. We hope that our solution will serve as a solid  baseline for future research on long-horizon manipulation tasks.
>
> **Regarding writing and figure sizes.**
>
> * Thank you for pointing out these issues. We will improve the writing in subsequent updates of the paper based on your comments. We improve the resolution and size of the robot execution frames in the figure (as shown in the uploaded pdf) and will add it to the revised paper.
>
> **Regarding Question 1.**
>
> * **The Sub-task Division:** At the macro level, the division of sub-tasks for long-horizon tasks is determined by analyzing the task's complexity and identifying logical, discrete components that can be learned independently. This division is typically guided by the natural segmentation of the task into sequential steps or phases, each requiring a different skill.
>
>   In our experimental setup, the division of sub-tasks was predefined according to the stages of task execution. For example, in the long-horizon task of "assembling a table," each sub-task was defined as "installing table leg 1" through "installing table leg 4." Similarly, in the "kitchen organization" task, the order of sub-tasks was predefined as "turn on the microwave," "move the kettle," "turn on the stove," and "turn on the light." The sub-tasks for each long-horizon task in our experiments are described in detail in Appendix C of the original paper.
>
> * **Impact of Different Sub-task Divisions:** Thank you for your insightful question. To address your query, we conducted experimental validation using the chair_ingolf task. The original sub-tasks in this task are divided as follows: "Assemble chair support 0 to target position" → "Assemble chair support 1 to target position" → "Assemble front leg 0 to target position" → "Assemble front leg 1 to target position." We have re-divided the sub-tasks into two alternative settings:
>
>   1. "Assemble chair support 0 and chair support 1 to target positions" → "Assemble front leg 0 to target position" → "Assemble front leg 1 to target position."
>   2. "Assemble chair support 0 to target position" → "Assemble chair support 1 to target position" → "Assemble front leg 0 and leg 1 to target positions."
>
>   Due to time constraints during the rebuttal phase, we were only able to run one test seed. The test results are as follows:
>
> * |      | chair_ingolf  (setup 1) | chair_ingolf  (setup 2) |
>   | :--: | :---------------------: | :---------------------: |
>   | SCaR |          0.68           |          0.74           |
>
> It is worth noting that, since the re-division of the sub-tasks results in only three sub-tasks, we set 90% as the success metric for all three sub-tasks being successfully executed. As seen in the table above, compared to SCaR's success rate of about 95% with the original four sub-task divisions, the success rate for completing the first sub-task and then executing the remaining two sub-tasks is significantly reduced. This decrease is due to the increased difficulty of the first sub-task in setup 1 (which requires assembling both chair support) and the last sub-task in setup 2. These changes result in a lower overall success rate for the task.
>
> This result suggests that a reasonable division of sub-tasks in long-horizon tasks is crucial for the success rate of overall task completion. We will add this experimental result and related discussion to an updated version of the paper.
>
> **Regarding Question 2.**
>
> * Thank you for the question. In the original experiment, we demonstrated the performance of SCaR for long-horizon manipulation tasks with up to 5 sub-tasks, specifically in the "Extended Kitchen" task, which includes the following sub-tasks:
>
>   1. Turn on the microwave
>   2. Turn on the stove
>   3. Turn on the light
>   4. Slide the cabinet to the right target position
>   5. Open the cabinet to the target position
>
>   To further answer your question, we added a sub-task to the Extended Kitchen task to evaluate SCaR's performance in manipulation tasks with longer horizons, involving 6 sub-tasks. The modified task, "Longer Extended Kitchen," includes:
>
>   1. Turn on the microwave
>   2. Turn on the stove
>   3. Turn on the light
>   4. Slide the cabinet to the right target position
>   5. Open the cabinet to the target position
>   6. **Move the kettle to the target position**
>
>   Due to the time constraints of the rebuttal period, we were only able to compare SCaR with T-STAR and run one test seed. The comparison test results are as follows:
>
> |        | Longer Extended kitchen |
> | :----: | :---------------------: |
> | T-STAR |          0.33           |
> |  SCaR  |          0.61           |
>
> As can be seen from the results, adding an additional sub-task increases the complexity and difficulty of the long-horizon task. Despite this,  SCaR still achieves a higher overall task execution success rate, with 28% higher compared to T-STAR. Although there is significant room  for further enhancement, we believe our solution offers a promising  baseline for future research in skill chaining methods for long-horizon  manipulation tasks. We will include this experimental result and related discussion in the updated version of the paper.

---

> > ### Comment · Reviewer_ysj9 · 2024-08-11
> >
> > Thank you for addressing my comments and questions. The additional experimental results are insightful and demonstrated the effectiveness of the method. Overall, the idea and method are well-motivated, and experiments are solid, although a harder real-world task would better demonstrate the usefulness of the algorithm. I have raised my score to weak accept.

---

> > > ### Author Response · Authors · 2024-08-11
> > >
> > > Thank you for your valuable suggestions! We promise to follow up on your comments by experimenting on harder real-world tasks to further validate the usefulness of the method, and will include relevant discussions with you in the revised version.

---

### Official Review · Reviewer_2GPv · 2024-07-13

**Soundness:** 3
**Presentation:** 3
**Contribution:** 2
**Rating:** 6
**Confidence:** 5

**Summary:**

In this work, the authors present SCaR or Skill Chaining via Dual Regularization, an algorithm for both learning a set of subtasks as well as how to sequence them well. The method consists of two major components, AES regularization for learning the skills from demonstrations and real world interaction, as well as a bi-directional adversatial regularization that ensures that there is a good overlap between one skill's ending states and another skill's beginning states.

The authors experimentally evaluate the method with experiments in two simulated environments and one real world setup. In the experiments, the proposed method outperform the baselines. The authors also show ablation experiments where they remove both the AES module and the bi-directionality module, and show that they both are important components.

**Strengths:**

+ The bi-directional regularization is a natural idea, and it is good to have confirmation that this works well in practice.
+ The bi-directional objective design also makes sense, and it is good to see that the natural setup works.
+ The experimental setup are interesting and and demonstrate the point of the authors re: the algorithm well.

**Weaknesses:**

- The AES component seems convoluted and overly complicated, and since it is completely unrelated to any of the skill chaining part, I am curious as to why all of the extra regularization was required.
- This point sticks out especially sorely because there are already good IL+RL methods out there, like [1], while it seems like the authors are struggling to fit a policy to their demo data+environment. I feel like I understand why this process is so difficult especially given that this has nothing to do with sequencing, and all about inverse RL.
- There are certain steps that seem particularly brittle, like equation 4, which is measuring progress, and it seems like it would be hard to scale to arbitrary environments with such a criteria.
- The setup seems very much directed towards setup with access to state. The simulated environments are both based on state, and the real robot experiment is also just learned in a simulation and zero-shot deployed in the real world (happy to be corrected if I am wrong). Something that can learn in the real world with image encodings as states would be much more robust.
- Training curves are not shown. In the appendix, the authors mention that they train sub-tasks for 150000000 (150 million!) steps, which seems practically out of the world to me as someone working on real robot. Especially given the existence of methods like [1], why are there so many steps needed to learn the sub-tasks?

[1] Haldar, Siddhant, et al. "Watch and match: Supercharging imitation with regularized optimal transport." Conference on Robot Learning. PMLR, 2023.

**Questions:**

- Why is the sub-task learning so slow even with demonstrations?
- Why use this particular method of learning from sub-tasks rather than using more efficient techniques?
- Could the discriminators be initialized by learning from the demo data?
- Are there works that talk about the Least-Squares GAIL more? The authors presented it without a citation, but it seems like it's more well known.

**Limitations:**

- The work relies on having access to states, which is hard in the real world and incredibly brittle.
- The number of training steps seems astronomically large, especially given demonstrations and for state-based environments.

---

> ### Author Rebuttal · Authors · 2024-08-06
>
> We sincerely thank you for acknowledging the strength of our work, e.g.,  the natural bi-directional regularization idea, interesting experimental setups, and effective experimental validation. We address the comments  and questions below.
>
> **Regarding the AES component.**
> - We introduced AES regularization in order to better pre-train the sub-task skills in the long-horizon task. Because our skill learning scheme combines RL and IL, this combined learning process not only allows the robot to explore the environment through the RL process to minimize possible suboptimality of learning from demonstrations alone, but also IL mitigates the over-exploration problem inherent in RL (as described in lines 44-48 of the original paper). The role of the AES, on the other hand, is to balance the RL and IL learning processes by monitoring the robot's skill learning process: if the robot struggles to imitate the expert demonstration effectively, it should focus more on self-learning from the environment. Conversely, if imitation is successful, it should continue to focus on the expert to improve the efficiency of sampling RL, as described in lines 156-158. This balance is consistent with how humans learn: if the teacher is ineffective, we learn independently; if the teacher is effective, we learn more from them. We will explain the AES regularity in more detail in an upcoming update of the paper.
>
> **Regarding the RL+IL setting.**
> - Please let us know if we have misunderstood your question. We will explain why we use the RL+IL setup and how our approach differs from the method mentioned in [1]. In the SCaR setting, the environment already includes a predefined reward function, allowing skills to be learned through agent interaction with the environment (RL) as well as learning from demonstrations (IL). In contrast, the approach in [1] requires inferring the reward function from demonstrations for IRL, which is fundamentally different from SCaR’s skill learning scheme.  We appreciate the reviewer's reminder, as we found that the reward inference process in ADS[2], which inspired our AES regularization, exploits the Optimal Transport concept from [1]. We will cite [1] and provide further discussion in the related work of future versions of this paper.
>
> **Regarding the imitation progress equation.**
> - The purpose of Equation 4 is to measure progress, primarily for use in AES regularization, to monitor the robot's learning process and to assess how well the robot imitates the expert. We have actually described this process in detail in Appendix B of the original paper. Admittedly, while we have mathematically modeled this process, it still has some limitations: this progress measure is only applicable to tasks where expert demonstrations are present in the environment and where trajectory similarity can be computed. We will discuss these limitations further in upcoming paper updates.
>
> **Regarding the state input.**
> - You are correct that SCaR does not currently handle encoded image inputs. We acknowledged this as a limitation and potential area for future development in the original paper. Our ongoing work is focused on enabling SCaR to process encoded image and semantic state inputs.
>
> **Regarding the training setting.**
> - We actually have shown the training curve in Fig.10 in the Appendix, where the x-axis represents the number of training steps and the y-axis shows the success rate of the sub-task. We chose 150M steps to train the sub-tasks to ensure that the skills are well-learned and  robust across various conditions and scenarios, which is critical for tasks that are content-rich and require high precision. And the benefits of high-quality policy learning in a simulated environment can often offset the computational costs, thus facilitating zero-shot  deployments in real-world applications.
>
> **Regarding Question 1.**
> - Thank you for your question. In fact, as shown in the training curves in Fig.10, our adaptive skill learning with AES regularization converges quickly and exhibits more stable training performance compared to other methods. In contrast, the comparison methods—PPO, GAIL, and fixed-RL-IL—do not achieve fast convergence or stable training performance, even with 150M training steps. This comparison highlights the  effectiveness of our proposed method.
>
> **Regarding Question 2.**
> - Thank you for your question. We chose the method of learning from sub-tasks because it is better suited to the long-horizon manipulation problem we are focusing on. The long-horizon tasks can be better accomplished by breaking them down into easier-to-learn sub-tasks, learning the corresponding skills for the sub-tasks, and then integrating the skills sequentially to execute the complete skill chaining. Learning the entire long-horizon task from scratch, without decomposing it into sub-tasks, using RL or IL is often challenging, as shown in the comparison results in Table 1 of the original paper.
>
> **Regarding Question 3.**
> - Please correct us if we misunderstand your question. If you are referring to the discriminator used in the GAIL part of IL in adaptive skill learning, the answer is yes. However, if you are referring to the bi-directional discriminator for the inter-skill chaining process, the answer is no. The initialization and learning of the bi-directional discriminator are not based on the demonstration data but instead rely solely on the set of successful initial and terminal states collected from the pre-trained skills.
>
> **Regarding Question 4.**
> - Thank you for pointing it out! The loss function for the least-squares discriminator was first proposed by [3], and we will include a citation for it in the updated version of the paper.
>
> [1] Haldar, Siddhant, et al. Watch and match: Supercharging  imitation with regularized optimal transport.
>
> [2] Liu, Yuyang, et al. Imitation Learning from Observation with Automatic Discount Scheduling.
>
> [2] X. Mao, et.al. Least Squares Generative Adversarial Networks.

---

> > ### Comment · Reviewer_2GPv · 2024-08-12
> >
> > Thank you for the rebuttal. I am strongly in favor of acceptance, and thus have increased my confidence score accordingly.

---

> > > ### Author Response · Authors · 2024-08-12
> > >
> > > Thank you very much for recognizing our work and for your suggestions! We promise to add the relevant discussion regarding your comments to the revised version.

---

### Official Review · Reviewer_uZgV · 2024-07-13

**Soundness:** 2
**Presentation:** 2
**Contribution:** 2
**Rating:** 6
**Confidence:** 4

**Summary:**

The paper proposes regularization strategies to enhance inter-skill and intra-skill consistency in skill chaining. Skill chaining is the problem of learning a sequence of sub-task policies chained together to achieve the goal, given expert demonstrations and rewards. The sub-task partition is defined manually before learning.

The paper builds its skill chaining framework on T-STAR [1]. Specifically:
- To regularize individual skill learning, the authors use Adaptive Equilibrium Scheduling (AES) strategy to adaptively balance the IL and RL objectives. The strategy is adapted from [2].
- To enhance inter-skill consistence, the authors extend the uni-directional reularization in [1] to bi-directional to push the initial set of successor skill and the termination set of previous skill close to each other.

The proposed method is evaluated on IKEA furniture assembly and Franka Kitchen benchmarks. It demonstrates improved performance against several skill chaining and RL / IL baselines, including T-STAR [1]. The paper further ablates on the two regularization strategies to demonstrate their necessity.

[1] Lee, Youngwoon, et al. "Adversarial Skill Chaining for Long-Horizon Robot Manipulation via Terminal State Regularization." 5th Annual Conference on Robot Learning.

[2] Liu, Yuyang, et al. "Imitation Learning from Observation with Automatic Discount Scheduling." The Twelfth International Conference on Learning Representations.

**Strengths:**

- Skill chaining is an important problem for robots to execute long-horizon tasks.
- The proposed regularization strategies are intuitive from a high-level.
- The experiments demonstrate the performance gain introduced by the proposed regularization strategies.

**Weaknesses:**

- While the skill chaining method outperforms baselines in experiments, it is merely a technical integration of existing methods.
  - The overall framework is built heavily on T-STAR [1]. It follows T-STAR exactly to formulate skill chaining as individual skill pretraining and joint fine-tuning, tackling skill pretraining with a combination of RL loss and GAIL loss, and aligning adjacent skills by training a discriminator.
  - The proposed AES strategy is adopted from [2].
  - The proposed bi-directional inter-skill regularization is merely a naive extension of the uni-directional one in [1].

- The explanation of proposed AES and bi-directional regularization doesn't seem to be clear enough for readers to understand without reading the related papers [1, 2].
  - In particular, I'm not sure how the bi-directional regularization works exactly. The key equation (6) of bi-directional regularization seems to be problematic - it learns a single classifier to classify both the initial set and termination set of the same skill.

**Questions:**

- I'm not sure how the bi-directional regularization works. Could you clarify on what are the classifiers trained for, and how they are trained (in equation 6)?

**Limitations:**

See Weakness

---

> ### Author Rebuttal · Authors · 2024-08-06
>
> We sincerely appreciate your constructive feedback and thank you for acknowledging the importance of the skill chaining, the high level of our proposed regularization strategies, and our experiments! We address your comments and questions below.
>
> **Regarding the reliance on existing methods:**
> - We appreciate the concerns raised by the reviewer but cannot fully agree with them. Our work goes beyond mere integration and introduces significant innovations and improvements. Below are the differences and innovations between SCaR and related methods:
>   - **Differences with T-STAR:** Our work focuses on skill-chaining approaches for long-horizon manipulation tasks. The basic premise is that sub-task skills for each phase of a long-horizon task must be pre-trained and then integrated, as discussed in the original paper(L27-29) and related work [1,3,4,5]. While our work draws on the T-STAR framework, T-STAR focuses only on the chaining process between skills. In contrast, SCaR enhances intra- and inter-skill dependencies by introducing Adaptive Equilibrium Scheduling (AES) Regularization and Bi-directional Regularization, two simple but effective schemes. Additionally, T-STAR uses a simple combination of RL loss and standard GAIL loss for skill pre-training, whereas SCaR incorporates a gradient penalty term in GAIL loss, improving training stability (Equation (3) of the original paper).
>   - **Differences with ADS:** Our proposed AES is inspired by ADS but fundamentally differs from it. ADS focuses on adjusting the discount factor during reinforcement learning training in Imitation Learning from Observation (ILfO), specifically adjusting $\gamma$ in $E_\pi[\sum_{t=1}^{T-1} \gamma^{t-1} r_t]$, which is essentially an inverse reinforcement learning (IRL) process.  Our proposed AES extends the concept of "discounted scheduling" in ADS to skill learning with both RL and IL processes, focusing on the scheduling of weight factors $\lambda_{\text{RL}}$ and $\lambda_{\text{IL}}$  for RL loss and GAIL loss around the sum of the environment feedback and predicted reward weights $\lambda_{\text{RL}} r_i^{\text{Env}}(s_t,a_t,s_{t+1},g) + \lambda_{\text{IL}} r^{\text{Pred}}_i(s_t,a_t; \phi)$, as described in Eq. (5) of the original paper.
>   - **Bi-directional and Uni-directional Regularization:** Bi-directional inter-skill regularization is a heuristic innovation we propose. While it may seem like a simple extension of uni-directional regularization, it effectively ensures better alignment and consistency between successive skills by simultaneously considering bi-directional constraints. Although simple, we hope our approach serves as a robust baseline for skill-chaining methods and informs future research on long-horizon manipulation tasks.
>
> **Regarding the clarity of methodology explanation.**
> * Thank you for pointing that out. We will include a more detailed  background on [1,2] in the appendix of the updated version to provide  readers with a better understanding of our methods.
>
> **Regarding the bi-directional regularization.**
> * To implement bi-directional regularization, we train a bi-directional discriminator, $\zeta_\omega$. We start by modeling each sub-task of the long-horizon task as a Markov Decision Process (MDP) and initialize 2-layer fully connected networks for each MDP as the initial dual discriminator $\zeta^k_\omega$. It is important to note that the dual discriminator is trained only after the corresponding skill policy has been pre-trained on each sub-task MDP. We initialize two buffers, $I_k$ and $\beta_k$, for each sub-task to store the initial and termination states of the skill success trajectory, respectively.
>
>   The pre-trained skill is then used to execute the corresponding sub-task and generate the trajectory. If sub-task $k$ is successfully completed, the initial state of the current trajectory is added to $I_k$, and the termination state is added to $\beta_k$.
>
>   Bi-directional regularization focuses on chaining at both ends, so we update the dual discriminator $\zeta^i_\omega$ using $\beta_{k-1}$, $I_k$, $\beta_k$, and $I_{k+1}$ after the trajectory of the $(k-1)$, $k$, and $(k+1)$th sub-task skills are executed. In our code implementation, we use two fully connected networks to minimize the training separately: one to make the terminal states of skill $k$ converge to the initial states of skill $k+1$, and the other to make the initial states of skill $k$ converge to the terminal states of skill $k-1$, as described in Equation (6) of the original paper. Finally, the parameters of the two networks are averaged and combined into $\zeta^k_\omega$.  We will further specify the different network subscripts in Eq. (6) to avoid confusion. Thank you for raising this issue.
>
>   Additionally, we have provided pseudo-code for bi-directional adversarial training in Algorithm 2 in Appendix A.2 of the original paper. We are currently refining our code to ensure clarity and ease of use. Once finalized, we commit to making it publicly accessible. We apologize for some typographical errors in the current Algorithm 2 (e.g., "dual set discriminator" should be "bi-directional discriminator"). We will correct these errors in the future update.
>
> **Regarding Question 1.**
> * Thank you for your question. Please refer to the response regarding **bi-directional regularization** for more details.
>
> [1] Lee, Youngwoon, et al.  Adversarial Skill Chaining for  Long-Horizon Robot Manipulation via Terminal State Regularization.
>
> [2] Liu, Yuyang, et al. Imitation Learning from Observation with  Automatic Discount Scheduling.
>
> [3] George Konidaris, et al.  Skill discovery in continuous reinforcement learning domains using skill chaining
>
> [4]  Lee, Youngwoon, et al.  Learning to coordinate manipulation skills via skill behavior diversification
>
> [5] Chen, Yuanpei, et al. Sequential dexterity: Chaining dexterous policies for long-horizon manipulation

---

> ### Comment · Reviewer_uZgV · 2024-08-10
>
> Thank you for clarifying on AES and bi-directional regularization! I'm glad to increase my score to Weak Accept, given the effectiveness of the proposed method. At the same time, I would further suggest the authors to:
>
> - Write the method section more clearly so the readers can understand more easily
> - Further improve on figure 1. As indicated by reviewer ysj912, it's still hard to see what happens in each frame. It also helps to label the sub-task names.
>
> Additionally: according to the authors' explanation, in Equation (6) $\mathcal{C}_2$, I suppose the first term should be $s_T \in \beta\_{i-1}$ and the second should be $s_I \in I_i$.

---

> ### Author Response · Authors · 2024-08-11
>
> Thank you for your insightful suggestions! We hereby make our commitment to write the method section more clearly based on your comments and to incorporate other minor changes discussed with all reviewers, such as improving the clarity of the figures, refining the subscripts of the equations, and adding more experiments, into the body of the revised paper.

---

### Official Review · Reviewer_9LLM · 2024-07-22

**Soundness:** 3
**Presentation:** 3
**Contribution:** 3
**Rating:** 7
**Confidence:** 3

**Summary:**

The paper introduces the Skill Chaining via Dual Regularization (SCaR) framework, designed to enhance skill chaining in long-horizon robotic manipulation tasks by applying dual regularization techniques during skill learning and chaining. The framework addresses the error accumulation issue in traditional skill chaining by focusing on both intra-skill and inter-skill dependencies, ensuring a more reliable execution over complex tasks like IKEA furniture assembly and kitchen organization. The effectiveness of SCaR is demonstrated through higher success rates and robustness in simulation and real-world tests compared to existing methods.

**Strengths:**

**Innovative Dual Regularization**: The dual regularization approach is a significant innovation that stabilizes the execution of chained skills by balancing intra-skill coherence and inter-skill alignment, addressing a common shortfall in existing methods.
 **Comprehensive Evaluation**: The paper provides extensive experimental results, including simulations and real-world tasks, which thoroughly demonstrate the effectiveness of SCaR compared to baseline methods.
 **Practical Impact**: The application to real-world tasks like furniture assembly and kitchen organization showcases the practical relevance and potential of SCaR in improving robotic automation efficiency and reliability.

**Weaknesses:**

**Limited Task Generalization Discussion**: The paper could expand on how SCaR adapts to varying task environments or tasks with different complexity levels, as most examples are confined to pre-defined scenarios with structured environments.
 **Dependency on Accurate Sub-task Definition**: The success of SCaR heavily relies on the precise definition and segmentation of sub-tasks, which might not be straightforward in more dynamic or less structured environments.  The paper mentions that the sub-task division is predefined and does not involve visual and semantic processing of objects. This could limit the framework's applicability in more complex, real-world scenarios where tasks might not be easily decomposable.

**Questions:**

1. How does SCaR handle scenarios where sub-tasks are not clearly defined or when unexpected sub-tasks arise due to changes in the environment?
2. Can you elaborate on the computational overhead introduced by the dual regularization process and its impact on real-time task execution?
3. Is there a strategy within SCaR to simplify the fine-tuning process, especially in adapting to different robots or task specifics without extensive retraining?

**Limitations:**

Although SCaR shows improvements in task execution, the scalability of this approach to even longer-horizon tasks involving more complex interactions and environmental variability remains less explored.

---

> ### Author Rebuttal · Authors · 2024-08-06
>
> We sincerely thank  you for acknowledging the strength of our work, e.g., the innovative dual regularization, comprehensive evaluation and practical impact. We address the comments and questions below.
>
> **Regarding the discussion of limited task generalizaiton.**
>
> * Thank you for pointing this out. The SCaR experimental setup in the original paper focused primarily on predefined structured environments. We chose these scenarios to validate the core mechanism and performance of SCaR, i.e., that it requires pre-training of predefined sub-task skills in a long-horizon task, and then stringing these skills together to perform the overall task. So the original paper does not discuss SCaR's ability to adapt to different task environments.
>
>   We have experimented and discussed the ability of SCaR to adapt in the face of unknown perturbations in the same environment. In Section 5.3 of the original paper, we tested the robustness of SCaR to perform a long line-of-sight task in the face of an unknown perturbation in that environment by adding an unexpected force to the robot arm joints during the execution phase of the sub-tasks of chair_bernhard and chair_ingolf. The experimental results show that our SCaR is robust to such perturbations compared to the comparison methods. More detailed experimental results are presented in Appendix D.2. We will discuss more about the limitations of SCaR's adaptability to different task environments in the next version of the paper in response to the reviewers' comments.
>
> **Regarding the dependency on accurate sub-task definition.**
>
> * Thank you for pointing this out. The current performance of SCaR does depend heavily on the rationality of sub-task division in long-horizon tasks, and we have done a simple experiment to illustrate this in our response to reviewer ysj9's question 1. Since the core innovation mechanism of our SCaR is mainly concerned with the learning of sub-task skills and skill chaining in long-horizon tasks, the sub-tasks of the long-horizon task in the experimental setting used in the current version of the paper are artificially divided and do not involve the model's ability to process visual and semantic objects.
>
>   We are actively exploring how to divide long-horizon tasks into sub-tasks without human intervention using large language models, and how to integrate visual and semantic processing techniques to extract key frames in the task to achieve a more rational division of sub-tasks that is more adaptable to real-world long-horizon tasks. We appreciate your insights and will discuss these future directions in the revised paper.
>
> **Regarding Question 1**.
>
> * Thank you for your question. Currently, SCaR's ability to optimize long-horizon task skill chaining is in the context of predefined sub-tasks, and we agree that enabling SCaR to handle scenarios where sub-tasks are not clearly defined or where unanticipated sub-tasks arise is an important direction to extend the boundaries of its capabilities.
>
>   We currently plan to utilize the capability of large models to extend the capability boundary of SCaR. Specifically, when facing long-horizon task scenarios with unclear or no sub-task definitions, we are trying to utilize the powerful task planning capability of the large model to divide reasonable sub-tasks for long-horizon tasks. When confronted with unexpected sub-tasks in the environment (e.g., the overall task goal is found to change), we are planning to utilize the large model for re-planning the sub-tasks. In this way, we will then train sub-task skills and realize skill chaining for long-horizon tasks through SCaR to extend its ability to handle various types of task scenarios.
>
>   We appreciate your feedback and will discuss these future directions in a revised paper to better elucidate this important aspect.
>
> **Regarding Question 2**.
>
> * Thank you for your question regarding computational overhead. We have actually reported the associated computational overhead and resource consumption in Appendix I.2 of the original paper.
>   The dual regularization process mainly increases the computational load of the offline pre-training phase of the skills, which takes about 10 hours per skill, while the pre-training of the skills without dual regularization (fixed-RL-IL) takes roughly more than 8 hours. The added computational overhead of the dual regularization process is within 2 hours.
>   And the process of SCaR chaining pre-trained skills (i.e., testing and evaluation) takes 10 to 15 hours, depending on the difficulty of the task. In the evaluation phase, we evaluated 5 seeds, each tested over 200 episodes, which translates to a real-time execution time of about 36-54 seconds for a single long-horizon task.
>   We will describe the computational overhead of each method at each stage in more detail in the updated version of the paper, and promise to make the code publicly available for further validation of computational efficiency after acceptance.
>
> **Regarding Question 3**.
>
> * Thank you for your question. To enable SCaR to adapt to different long-horizon manipulation tasks or robots (such as assembling a four-legged table, assembling a chair with three components, or organizing a kitchen), it indeed requires retraining sub-task skills.
>  When SCaR needs to adapt to different scenarios within the same task distribution, such as assembling a four-legged table where only the table’s design changes, we believe that integrating techniques like online learning into the skill learning process can help. By using online gradient descent to gradually update model parameters, we can fine-tune the skills for assembling tables of different designs without extensive retraining, allowing for rapid task adaptation.
> Thank you again for your insightful comments. We will incorporate this discussion and follow-up work plan into the updated paper.

---

> > ### Comment · Reviewer_9LLM · 2024-08-12
> > **Thanks for your reply**
> >
> > Thank you for the rebuttal. I am strongly in favor of acceptance.

---

> > > ### Author Response · Authors · 2024-08-13
> > >
> > > Thank you very much for recognizing our work and for your constructive suggestions! We promise to include relevant discussion regarding your comments in the revised version.

---

### Author Rebuttal · Authors · 2024-08-07

**Global response**

Dear Reviewers,

Thank you very much again for your helpful comments. We appreciate your recognization of our work and would like to engage with you in our responses to your questions/comments. We hope that our approach will provide a simple and solid baseline for future research on skill chaining methods for long-horizon manipulation tasks.

 If you have any questions about our work or our response, we would be happy to discuss them further.

Best Regards,

the Authors.

---

### Decision · Program_Chairs · 2024-09-25

**Decision:**

Accept (poster)

**Comment:**

This paper proposes an advanced skill chaining algorithm, addressing long-horizon tasks. While the proposed method builds upon the existing T-STAR framework, it introduces two key innovations: adaptive equilibrium scheduling and bi-directional regularization, which significantly improve the robustness of skill chaining. The empirical evaluations are comprehensive, demonstrating superior performance on complex tasks, such as furniture assembly and kitchen organization.

The reviewers initially had concerns, but after the rebuttal phase, they reached a consensus to accept this paper, contingent on additional experimental results and clarifications regarding the methodology and experimental details.

The authors have committed to enhance the paper by (1) improving the overall writing and figure quality, (2) incorporating discussion points, particularly regarding the sub-task decomposition, (3) addressing more challenging tasks in both simulated and real-world environments.